# CO₂-equivalence metrics for surface albedo change based on the radiative forcing concept: A critical review

Ryan M. Bright[1] and Marianne T. Lund[2]

[1]Department of Forest and Climate, Norwegian Institute of Bioeconomy Research (NIBIO), PO Box 115, 1431-Ås, Norway
[2]Centre for International Climate Research (CICERO), 0349 Oslo, Norway

*Correspondence to*: Ryan M. Bright (ryan.bright@nibio.no)

**Abstract.** Management of Earth's surface albedo is increasingly viewed as an important climate change mitigation strategy both on (Seneviratne et al., 2018) and off (Field et al., 2018;Kravitz et al., 2018) the land. Assessing the impact of a surface albedo change involves employing a measure like radiative forcing (*RF*) which can be challenging to digest for decision-makers who deal in the currency of $CO_2$-equivalent emissions. As a result, many researchers express albedo change (*Δα*) *RF*s in terms of their $CO_2$-equivalent effects, despite the lack of a standard method for doing so, such as there is for emissions of well-mixed greenhouse gases (WMGHGs; e.g., IPCC AR5, Myhre et al. (2013)). A major challenge for converting *Δα RF*s into their "$CO_2$-equivalent" effects in a manner consistent with current IPCC emission metric approaches stems from the lack of a universal time-dependency following the perturbation (perturbation "lifetime"). Here, we review existing methodologies based on the *RF* concept with the goal of highlighting the context(s) in which the resulting $CO_2$-equivalent metrics may or may not have merit. To our knowledge this is the first review dedicated entirely to the topic since the first $CO_2$-eq. metric for *Δα* surfaced 20 years ago. We find that, although there are some methods that sufficiently address the time-dependency issue, none address or sufficiently account for the spatial disparity between the climate response to $CO_2$ emissions and *Δα* – a major critique of *Δα* metrics based on the *RF* concept (Jones et al., 2013). We conclude that considerable research efforts are needed to build consensus surrounding the *RF* "efficacy" of various surface forcing types associated with *Δα* (e.g., crop change, forest harvest, etc.), and the degree to which these are sensitive to the spatial pattern, extent, and magnitude of the underlying surface forcings.

## 1 Introduction

The albedo at Earth's surface helps to govern the amount of solar energy absorbed by the earth system and is thus a relevant physical property shaping weather and climate (Cess, 1978;Hansen et al., 1984;Pielke Sr. et al., 1998). On average, Earth reflects about 30% of the energy it receives from the sun, of which about 13% may be attributed to the surface albedo (Stephens et al., 2015;Donohoe and Battisti, 2011). In recent years it has become the subject of increasing research interest amongst the scientific community, as measures to increase Earth's surface albedo are increasingly viewed as an integral component of climate change mitigation and adaptation, both on (Seneviratne et al., 2018) and off (Field et al., 2018;Kravitz et al., 2018) the

land. Surface albedo modifications associated with large-scale carbon dioxide removal (CDR) like re-/afforestation can detract from the effectiveness of such mitigation strategies (Boysen et al., 2016), given that such modifications generally serve to increase Earth's solar radiation budget resulting in warming. Like emissions of GHGs and aerosols, perturbations to the planetary albedo via perturbations to the surface albedo represent true external forcings of the climate system and can be measured in terms of changes to Earth's radiative balance – or radiative forcings (Houghton et al., 1995). The radiative forcing

(*RF*) concept provides a first-order means to compare surface albedo changes (henceforth $\Delta\alpha$) to other perturbation types, thus enabling a more comprehensive evaluation of human activities altering Earth's surface (Houghton et al., 1995;Pielke Sr. et al., 2002).

*Radiative forcing is* a standard measure of the effects of various emissions or perturbations on climate, and can be used to
compare the effect of changes between any two points in time. It isa backward-looking measure accounting for the impact up to the given point anddoes not express the actual temperature response to the perturbation. To enable aggregation of emissions of different gases to a common scale, the concept of $CO_2$-equivalent emissions is commonly used in assessments, decision making, and policy frameworks. While initially introduced to illustrate the difficulties related to comparing the climate impacts of different gases, the field of emission metrics -- i.e., the methods to convert non-$CO_2$ radiative constituents into their $CO_2$-
equivalent effects – has evolved and presently includes a suite of alternative formulations, including the Global Warming Potential (GWP) adopted by the UNFCCC (O'Neill, 2000;Fuglestvedt et al., 2003;Fuglestvedt et al., 2010). Today, $CO_2$-equivalency metrics form an integral part of UNFCC emission reporting and climate agreements (e.g. The Kyoto Protocol) – in addition to the fields of Life Cycle Assessment (Heijungs and Guineév, 2012) and Integrated Assessment Modeling (O'Neill et al., 2016) – despite much debate around GWP as the metric of choice (Denison et al., 2019). As such, many researchers
seek to convert *RF* from $\Delta\alpha$ into a $CO_2$-equivalent effect, which is particularly useful in land use forcing research when perturbations to terrestrial carbon cycling often accompany the $\Delta\alpha$. Although seemingly straight-forward at the surface, the procedure is complicated by two key fundamental differences between $\Delta\alpha$ and $CO_2$: additional $CO_2$ becomes well-mixed within the atmosphere upon emission, and, the resulting atmospheric perturbation persists over millennia and cannot be fully reversed by human interventions. In other words, $CO_2$'s *RF* is both temporally- and spatially-extensive with the ensuring
climate response being independent of the location of emission, whereas the *RF* and ensuing climate response following $\Delta\alpha$ is more localized and can be fully reversed on short time scales.

These challenges have led researchers to adapt a variety of diverging methods for converting albedo-change *RF*s (henceforth $RF_{\Delta\alpha}$) into $CO_2$-equivalence. Unlike for conventional GHGs, however, there has been little concerted effort by the climate
metric science community to build consensus or formalize a standard methodology for $RF_{\Delta\alpha}$ (as evidenced by IPCC AR4 and AR5). Here, we review existing $CO_2$-equivalent metrics for $\Delta\alpha$ and their underlying methods based on the *RF* concept. To our knowledge this is the first review dedicated to the topic since the first $\Delta\alpha$ metric surfaced 20 years ago. Herein, we compare and contrast existing metrics both quantitatively and qualitatively, with the main goal of providing added clarity surrounding

the context in which the proposed metrics have (de)merits.  We start in Section 2 by providing an overview of the methods conventionally applied in the climate metric context for estimating radiative forcings following $CO_2$ emissions and surface albedo change.  We then present the reviewed $\Delta\alpha$ metrics in Section 3 and systematically evaluate them quantitatively in Section 4 and qualitatively in Section 5.  In Section 6 we review and evaluate a relatively new usage of the GWP metric previously unapplied as a $\Delta\alpha$ metric – termed *GWP\** -- while in Section 7 we review the interpretation challenges of a $CO_2$-eq. measure for $\Delta\alpha$ based on the *RF* concept.  We conclude in Section 8 with a discussion about the limitations and uncertainties of the reviewed metrics, while providing recommendations and guidance for future application.

## 2 Radiative forcings from CO₂ emissions and surface albedo change

IPCC emission metrics are based on the stratospherically-adjusted *RF* at the tropopause in which the stratosphere is allowed to relax to the thermal steady state (Myhre et al., 2013;IPCC, 2001).  Estimates of the stratospheric *RF* for $CO_2$ (henceforth $RF_{CO_2}$) are derived from atmospheric concentration changes imposed in global radiative transfer models (Myhre et al., 1998;Etminan et al., 2016).  For shortwave *RF*s there is no evidence to suggest that the stratospheric temperature adjusts to a surface albedo change (at least for land-use and land cover change (LULCC), Smith et al. (2020); Hansen et al., (2005); Huang et al. (2020)) and thus the instantaneous shortwave flux change at TOA is typically taken as $RF_{\Delta\alpha}$ , consistent with Myhre et al. (2013).

One of the major critiques of the instantaneous or stratospherically-adjusted *RF* is that they may be inadequate as predictors of the climate response (i.e., changes to near surface air temperatures, precipitation, etc.).  The climate may respond differently to different perturbation types despite similar *RF* magnitudes – or in other words – feedbacks are not independent of the perturbation type (Hansen et al., 1997;Joshi et al., 2003).  Alternative *RF* definitions that include tropospheric adjustments (Shine et al., 2003) or even land surface temperature adjustments (Hansen et al., 2005) have been proposed with the argument that such adjustments are more indicative of the type and magnitude of feedbacks underlying the climate response (Sherwood et al., 2015;Myhre et al., 2013).   These alternatives – referred to as "effective radiative forcings (*ERF*)" – may be preferred when they differ notably from the instantaneous- or stratospherically-adjusted *RF*, in which case their use might be preferred in metric calculations.  Alternatively, climate "efficacies" can be applied to adjust instantaneous or stratospherically-adjusted *RF* – where efficacy is defined as the temperature response to some perturbation type relative to that of $CO_2$.  The implications of applying efficacies for spatially heterogenous perturbations like $\Delta\alpha$ are discussed further in Section 6.

## 2.1 CO₂ radiative forcings

Simplified expressions for the global mean $RF_{CO_2}$ (in W m⁻²) due to a perturbation to the atmospheric $CO_2$ concentration are based on curve fits of radiative transfer model outputs (Myhre et al., 1998;Myhre et al., 2013) :

$$RF_{CO_2}(\Delta C) = 5.35 \, ln\left(\frac{C_0 + \Delta C}{C_0}\right) \tag{1}$$

where $C_0$ is the initial concentration and $\Delta C$ is the concentration change. Because of the logarithmic relationship between $RF$ and $CO_2$ concentration, $CO_2$'s radiative efficiency – or the radiative forcing per unit change in concentration over a given background concentration – decreases with increasing background concentrations. When $\Delta C$ is 1 ppm and $C_0$ is the current

concentration, we may then refer to the solution of Eq. (1) as $CO_2$'s current global mean radiative efficiency – or $\alpha_{CO_2}$ (in W m$^{-2}$ ppm$^{-1}$).

Updates to the $RF_{CO_2}$ function (Eq. 1) were given in Etminan et al. (2016) where the constant 5.35 (or $RF_{2\times CO_2}$/ln[2]) was replaced by an explicit function of $CO_2$, $CH_4$, and $N_2O$ concentrations. However, this update is only important for very large

$CO_2$ perturbations and is unnecessary to consider for emission metrics that utilize radiative efficiencies for small perturbations around present-day concentrations (Etminan et al., 2016).

For emission metrics, it is more convenient to express $CO_2$'s radiative efficiency in terms a mass-based concentration increase:

$$k_{CO_2} = \frac{\alpha_{CO_2} \varepsilon_{air} 10^6}{\varepsilon_{CO_2} M_{atm}} \tag{2}$$

where $\alpha_{CO_2}$ is the radiative efficiency per 1 ppm concentration increase, $\varepsilon_{CO_2}$ is the molecular weight of $CO_2$ (44.01 kg kmol$^{-1}$), $\varepsilon_{air}$ is the molecular weight of air (28.97 kg kmol$^{-1}$), and $M_{atm}$ is the mass of the atmosphere ($5.14 \times 10^{18}$ kg). The solution of Eq. (2) thus yields $CO_2$'s global mean radiative efficiency with units in W m$^{-2}$ kg$^{-1}$.


The global mean radiative forcing over time following a 1 kg pulse emission of $CO_2$ can be estimated with an impulse-response function describing atmospheric $CO_2$ removal in time by Earth's ocean and terrestrial CO2 sinks:

$$RF_{CO_2}(t) = k_{CO_2} \int_{t=0}^{t} y_{CO_2}(t)dt \tag{3}$$


where $y_{CO_2}$ is a model describing the decay of $CO_2$ in the atmosphere over time. In AR5 $y_{CO_2}$ is based on the multi-model mean $CO_2$ impulse-response function described in (Joos et al., 2013;Myhre et al., 2013) for a $CO_2$ background concentration of 389 ppmv, $t$ is the time step, and $k_{CO_2}$ is the radiative efficiency per kg $CO_2$ emitted upon the same background concentration (i.e., $1.76 \times 10^{-15}$ W m$^{-2}$ kg$^{-1}$) which is assumed constant and time-invariant for small perturbations and for the calculation of

emission metrics (Joos et al., 2013;Myhre et al., 2013). The pulse-response function ($y_{CO_2}$) comprises four carbon pools representing the combined effect of several carbon-cycle mechanisms rather than directly corresponding to individual physical

processes. Although considered ideal for metric calculations in IPCC AR5, state-dependent alternatives exist in which the carbon cycle response is affected by rising temperature or $CO_2$ accumulation in the atmosphere (Millar et al., 2017).

For an emission (or removal) scenario, $RF_{CO_2}(t)$ is estimated from changes to atmospheric $CO_2$ abundance computed as a convolution integral between emissions (or removals) and the $CO_2$ impulse-response function:

$$RF_{CO_2}(t) = k_{CO_2} \int_{t'=0}^{t} e(t') y_{CO_2}(t - t') dt' \tag{4}$$

where $t$ is the time dimension, $t'$ is the integration variable, $e(t')$ is the $CO_2$ emission (or removal) rate (in kg).

## 2.2 Shortwave radiative forcings from surface albedo change

The time step of Eq. (3) is typically one year, thus it is convenient to utilize an annually averaged $RF_{\Delta\alpha}$ when deriving a $CO_2$-equivalent metric. Given the asymmetry between solar irradiance and the seasonal cycle of surface albedo in many extra-tropical regions, a more precise estimate of the annual $RF_{\Delta\alpha}$ is one based on the monthly (or even daily) $\Delta\alpha$ (Bernier et al.,
140 2011).

The local annual mean instantaneous $RF_{\Delta\alpha}$ (in W m$^{-2}$) following monthly surface albedo changes (unitless) can be estimated with radiative kernels derived from global climate models [e.g., (Soden et al., 2008;Pendergrass et al., 2018;Block and Mauritsen, 2014;Smith et al., 2018)], although it should be pointed out that kernels are model- and state-dependent. Bright &
O'Halloran (2019) recently presented a simplified $RF_{\Delta\alpha}$ model allowing greater flexibility surrounding the prescribed atmospheric state, given as:

$$RF_{\Delta\alpha}(t) = \frac{1}{12} \sum_{m=1}^{12} -SW_{\downarrow,m,t}^{sfc} \sqrt{T_{m,t}} \Delta\alpha_{m,t} \tag{5}$$

where $\Delta\alpha_{m,t}$ is a surface albedo change in month $m$ and year $t$, $SW_{\downarrow}^{sfc}$ is the incoming solar radiation flux incident at surface level in month $m$ and year $t$, and $T_{m,t}$ is the all-sky monthly mean clearness index (or $SW_{\downarrow}^{sfc}/SW_{\downarrow}^{toa}$ ; unitless) in month $m$ and year $t$.

It is important to reiterate that the $RF_{\Delta\alpha}$ defined with either Eq. (5) or GCM-based kernels strictly represents the instantaneous
shortwave flux change at TOA and is not directly comparable to other definitions of $RF$ based on net (downward) radiative flux changes at TOA following atmospheric adjustments. A perturbation to $\Delta\alpha$ will result in a modification to the turbulent heat fluxes leading to radiative adjustments in the troposphere (Laguë et al., 2019;Huang et al., 2020;Chen and Dirmeyer,

2020). However, in the context of emission metrics, both $RF_{\Delta\alpha}$ and $RF_{CO_2}$ have merit given that they do not require coupled climate model runs of several years to compute.

## 3 Overview of CO₂-equivalent metrics for $RF_{\Delta\alpha}$

Over the past 20 years, a variety of metrics and their permutations have been employed to express $RF_{\Delta\alpha}$ as "CO₂-equivalence", as evidenced from the 27 studies included in this review (Table 1).

**Table 1.** Studies included in this review.

| Study | Metric | Notes |
|-------|--------|-------|
| Betts (2000) | *EESF* | AF = 0.5 |
| Akbari et al. (2009) | *EESF* | AF = 0.55 |
| Montenegro et al. (2009) | *EESF* | AF = 0.5 |
| Thompson et al. (2009a) | *EESF* | AF = 0.5 |
| Thompson et al. (2009b) | *EESF* | AF = 0.5 |
| Muñoz et al. (2010) | *EESF* | AF based on C-cycle model and *TH* = 20, 100, and 500 yrs. |
| Menon et al. (2010) | *EESF* | AF = 0.55 |
| Georgscu et al. (2011) | *EESF* | AF = 0.50 |
| Cherubini et al. (2012) | *GWP* | Based on effective *RF* estimated with a climate efficacy of 1.94[b] |
| Bright et al. (2012) | *GWP* | TH = 20; 100; 500 yrs. |
| Susca, T. (2012b) | $\sum$*TDEE*[a] | |
| Susca, T. (2012a) | $\sum$*TDEE*[a] | |
| Guest et al. (2013) | *GWP* | |
| Zhao & Jackson (2014) | *EESF* | AF = 0.5; Based on effective *RF* estimated with a climate efficacy of 0.52[c] |
| Caiazzo et al. (2014) | *EESF* | AF based on C-cycle model and *TH* = 100 yrs. |
| Singh et al. (2014) | *GWP* | TH = 100 yrs. |
| Bright et al. (2016) | *TDEE*; $\sum$*TDEE* | |
| Mykleby et al. (2017) | *EESF* | AF based on C-cycle model and *TH* = 80 yrs. |
| Fortier et al. (2017) | *EESF* | AF based on C-cycle model and *TH* = 100 yrs. |
| Carrer et al. (2018) | *EESF/TH* | AF based on C-cycle model and *TH* = 100 yrs. |
| Carrer et al. (2018) | *GWP/TH* | TH = 100 yrs. |

| | | |
|---|---|---|
| Favero et al. (2018) | *EESF* | AF based on C-cycle model and *TH* = 100 yrs. |
| Sieber et al. (2019) | *GWP* | TH = 100 yrs. |
| Sieber et al. (2020) | *GWP* | TH = 100 yrs. |
| Genesio et al. (2020) | *EESF* | AF = 0.47 |
| Sciusco et al. (2020) | *EESF/TH* | AF based on C-cycle model and *TH* = 100 yrs. |
| Bright et al. (2020) | *TDEE*; | |
| | $\sum TDEE$ | |
| Lugato et al. (2020) | *GWP* | TH = 84 yrs. |

[a] Referred to as "time-dependent emission"; [b] From idealized climate model simulations of high Arctic snow albedo change (Bellouin and Boucher, 2010); [c] From idealized climate model simulations of global LULCC (Davin et al., 2007)

Chiefly differentiating the methods behind the metrics shown in Table 1 – described henceforth – is how time is represented with respect to both the $\Delta\alpha$ and the reference gas (i.e., CO2) perturbations.  Among the most common approaches is to relate

$RF_{\Delta\alpha}$ to the $RF$ following a $CO_2$ emission imposed on some atmospheric $CO_2$ concentration background, but with a fraction of the emission instantaneously removed by Earth's ocean and terrestrial $CO_2$ sinks by an amount defined by one minus the so-called "airborne fraction" (AF) – or the growth in atmospheric $CO_2$ relative to anthropogenic $CO_2$ emissions (Forster et al., 2007).

This method – or the "emissions equivalent of shortwave forcing (*EESF*)" – was first introduced by Betts (2000) and may be expressed (in kg $CO_2$-eq. m$^{-2}$) as:

$$EESF = \frac{RF_{\Delta\alpha}}{k_{CO_2} A_E AF} \tag{6}$$

where $RF_{\Delta\alpha}$ is the local annual mean instantaneous $RF$ from a prescribed monthly $\Delta\alpha$ scenario  (in W m$^{-2}$), $k_{CO_2}$ is the global mean radiative efficiency of $CO_2$ (e.g., Eq. (2); in W m$^{-2}$ kg$^{-1}$),  $A_E$ is Earth's surface area ($5.1 \times 10^{14}$ m$^2$), and AF is the airborne fraction.  Because AF appears in the denominator in Eq. (6), the $CO_2$ equivalent estimate will be highly sensitive to the choice of AF.  Figure 1 plots AF since 1959 which, as can be seen, can fluctuate considerably over short time periods, ranging from a high of 0.81 in 1987 and low of 0.20 in 1992.

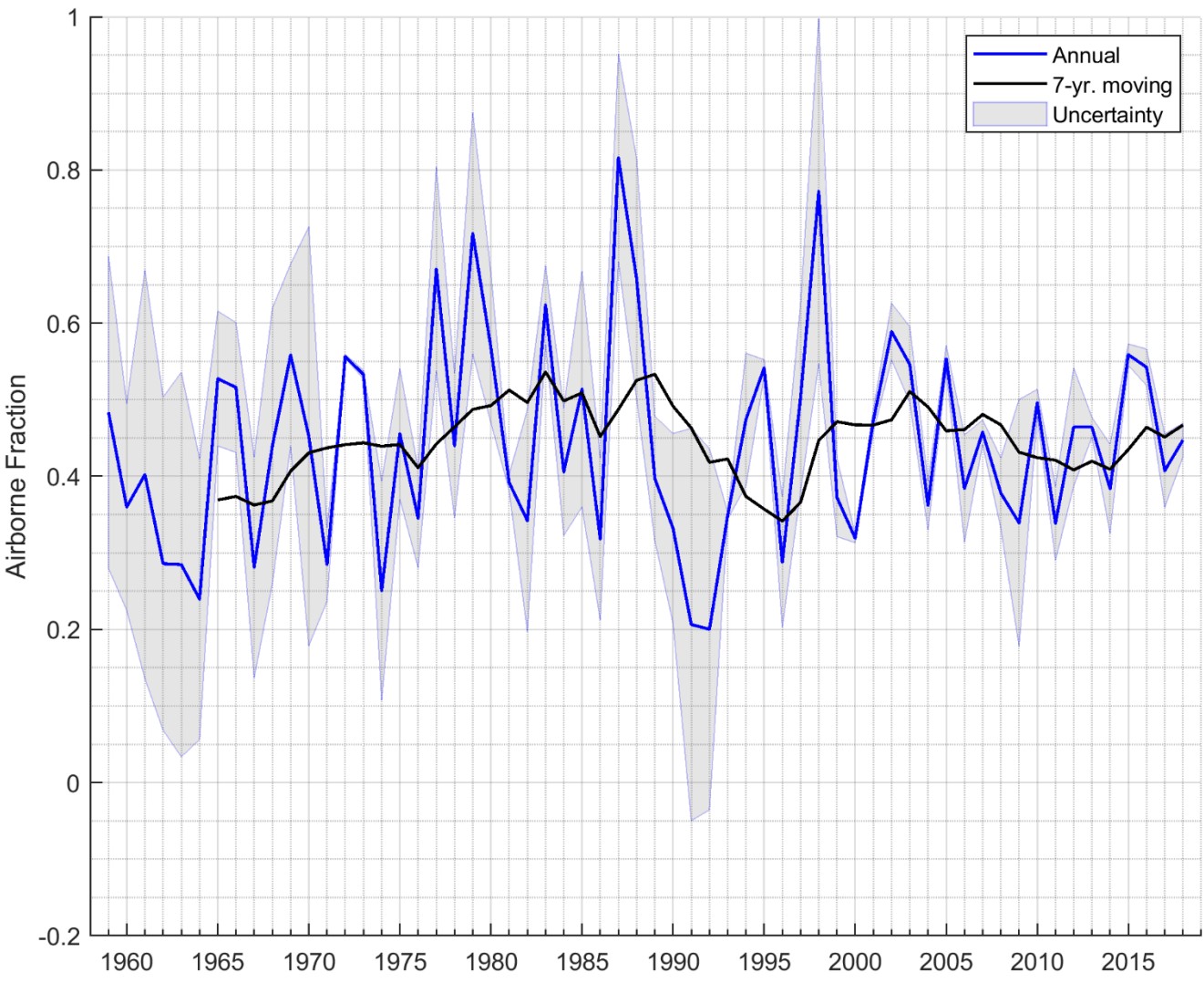

**Figure 1. 1959-2018 airborne fraction (*AF*), defined here as the growth in atmospheric CO₂ – or the atmospheric CO₂ remaining after removals by ocean and terrestrial sinks – relative to anthropogenic CO₂ emissions (fossil fuels and LULCC). "Uncertainty" is defined as $AF \pm |BI|/E$, where $E$ are total anthropogenic CO₂ emissions and $BI$ is the budget imbalance – or $E$ minus the sum of atmospheric CO₂ growth and CO₂ sinks. Underlying data are from the Global Carbon Project (Friedlingstein et al., 2019).**

More importantly, use of AF in Eq. (6) means that time-dependent atmospheric $CO_2$ removal processes following emissions are not explicitly represented. However, using the AF may be justifiable in some contexts – such as when $\Delta\alpha$ has no time dependency (on inter-annual scales). For example, the pioneering study by Betts (2000) – to which almost all $CO_2$-eq. literature for $\Delta\alpha$ may be traced (Table 1) – made use of AF when estimating $CO_2$-equivalence of $RF_{\Delta\alpha}$ because the research objective was to compare an albedo contrast between a fully grown forest and a cropland (i.e, $\Delta\alpha$) to the stock of $CO_2$ in the forest – a stock that had been assumed to accumulate over 80 yrs. which is the approximate time frame over which Earth's $CO_2$ sinks

function to remove atmospheric $CO_2$ to a level conveniently represented by the chosen AF. Had a transient or interannual $\Delta\alpha$ scenario been modeled, however, applying the *EESF* method at each time step of the scenario would have severely overestimated $CO_2$-equivalent emissions.


For this reason, Bright *et al.* (2016) argued that for time-dependent $\Delta\alpha$ scenarios (i.e., when $\Delta\alpha$ evolves over interannual time scales), the time-dependency of $CO_2$ removal processes (atmospheric decay) following emissions should be taken explicitly into account when estimating the effect characterized in terms of $CO_2$-equivalent emissions (or removals), thus proposing an alternate metric termed "time-dependent emissions equivalence" -- or *TDEE*:


$$TDEE = A_E^{-1} k_{CO_2}^{-1} Y_{CO_2}^{-1} RF_{\Delta\alpha}^* \qquad (7)$$

where *TDEE* is a column vector of $CO_2$-equivalent emission (or removal) pulses (i.e., one-offs) with length defined by the number of time steps (e.g., years) included in the $\Delta\alpha$ time series (in kg $CO_2$-eq. m$^{-2}$ yr$^{-1}$), $RF_{\Delta\alpha}^*$ is a column vector of the local
annual mean instantaneous $RF_{\Delta\alpha}$ (in W m$^{-2}$) corresponding to the $\Delta\alpha$ time series (or $RF_{\Delta\alpha}(t)$), and $Y_{CO_2}$ is a lower triangular matrix with column (row) elements as the atmospheric $CO_2$ fraction decreasing (increasing) with time (i.e., $y_{CO_2}(t)$). The elements in vector *TDEE* thus give the $CO_2$-equivalent series of emission (or removal) pulses in time yielding the instantaneous $RF_{\Delta\alpha}$ time profile ($RF_{\Delta\alpha}(t)$) corresponding to the temporally-explicit $\Delta\alpha$ scenario ($\Delta\alpha(t)$). Summing all elements in *TDEE* (i.e., $\sum TDEE$) gives a measure of the accumulated $CO_2$-eq. emissions (removals) over time. The *TDEE* approach is
conceptually similar to the $CO_2$-forcing-equivalence ("$CO_2$-fe") approach (Jenkins et al., 2018;Zickfeld et al., 2009) building on the notion of a "forcing equivalent" index (Wigley, 1998).

Time-dependent metrics like the well-known Global Warming Potential (*GWP*) (Shine et al., 1990;Rogers and Stephens, 1988) have also been applied to characterize $\Delta\alpha(t)$, which accumulates $RF_{\Delta\alpha}(t)$ over time (temporally-discretized) up to some
policy or metric time horizon (*TH*) which is then normalized to the temporally-accumulated radiative forcing following a unit pulse $CO_2$ emission over the same *TH*:

$$GWP_{\Delta\alpha}(TH) = \frac{\sum_0^{t=TH} RF_{\Delta\alpha}(t)}{A_E k_{CO_2} \sum_0^{t=TH} y_{CO_2}(t)} \qquad (8)$$

where *TH* is the temporal accumulation or metric time horizon. Because it is a cumulative measure, studies making use of *GWP* often divide by the number of time steps (*TH*) to approximate an annual $CO_2$ flux (e.g., Carrer et al., (2018)). The result of Eq. (8) can be interpreted an equivalentpulse of $CO_2$ (in kg $CO_2$-eq. m$^{-2}$) at $t = 0$ giving the same time-integrated *RF* at *TH* as that following a 1 kg pulse of $CO_2$.

## 3.1 Metric permutations

Some studies have applied various permutations of the three metrics presented above. For instance, some have applied definitions of the airborne fraction (AF) based on $CO_2$'s pulse-response function (i.e., $y_{CO_2}(t)$) when estimating *EESF* on the grounds that the analysis required a long and forward-looking time perspective (Caiazzo et al., 2014;Favero et al., 2018;Mykleby et al., 2017;Muñoz et al., 2010;Sciusco et al., 2020). A consequence is that the magnitude of the $CO_2$-eq. calculation is highly sensitive to the subjective choice of the *TH* chosen as the basis for the AF (typically taken as the mean atmospheric fraction for the period up to $TH$ – or $TH^{-1}\int_{t=0}^{t=TH} y_{CO_2}(t)\,dt$). Other permutations include the normalization of *EESF* or *GWP(TH)* by *TH* to arrive at a uniform time series of $CO_2$-eq. pulses (Carrer et al., 2018), or the summing of *TDEE* up to *TH* to obtain a $CO_2$-eq. stock perturbation measure (Bright et al., 2020;Bright et al., 2016).

## 3.2 Metric decision tree

Their relative merits and drawbacks (further discussed in Sections 4 & 5) notwithstanding, Figure 2 presents a decision tree for differentiating between the reviewed *Δα* metrics presented heretofore.

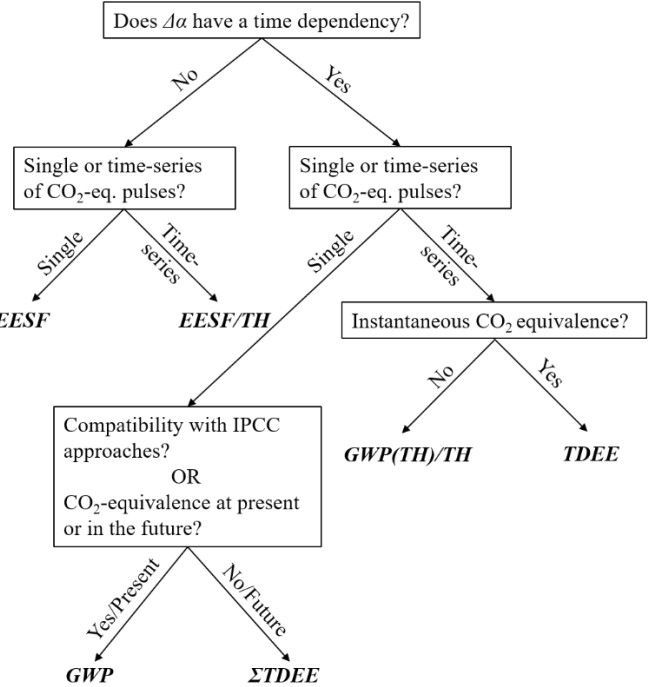

**Figure 2. Decision-tree for metrics applied in the literature included in this review to assess *Δα*.**

A principle differentiator after the time-dependency distinction is whether $CO_2$-equivalence corresponds to a single emission (removal) pulse or a time series of multiple $CO_2$-equivalent emission (removal) pulses. For the time-dependent metrics (Fig. 2, right branch), further distinction can be made according to whether the $CO_2$-equivalent effect is an instantaneous effect (in the case of the time-series measures) and whether IPCC compatibility is desired by the practitioner (in the case of the single pulse measures). By "IPCC compatibility", we mean that the metric computation and physical interpretation aligns with

emission metrics presented in previous IPCC climate assessment reports and IPCC good practice guidelines for national emission inventory reporting. A second or alternate distinction can be made for the time-dependent and single pulse measures according to whether the $CO_2$-equivalent effect corresponds to the present ($t = 0$) or the future ($t = TH$).

### 3.3 Δα vs. emission metrics

All metric application entails subjective user decisions, such as type of metric (i.e., instantaneous vs. accumulative; scalar vs.

time-series) and time horizon for impact evaluation. $CO_2$-eq. metrics for $\Delta\alpha$ require additional decisions by the practitioner affecting both their transparency and uncertainty, which are highlighted in Table 3.

**Table 3. Important decisions required by the practitioner to obtain a $CO_2$-eq. metric for $\Delta\alpha$ (based on *RF*) relative to conventional $CO_2$-normalized emission metrics of the IPCC (i.e., *GWP*).**

| Radiative forcing agent | *RF* Metric | Initial Perturbation (emission or $\Delta\alpha$) | Perturbation time-dependency | *RF* model |
|---|---|---|---|---|
| GHGs | *GWP* | Unit pulse | IPCC | IPCC |
| $\Delta\alpha$, time-dependent | *TDEE; GWP* | User defined | User-defined | User defined |
| $\Delta\alpha$, time-independent | *EESF* | User defined | None | User defined |


First among these is the need to quantify the initial physical perturbation (i.e., $\Delta\alpha$) which is irrelevant for IPCC emission metrics where the initial perturbation is a unit pulse emission. For $\Delta\alpha$ metrics, uncertainty surrounding estimates of the initial (or reference) and perturbed albedo states is introduced. Second, for the time-dependent metrics (Table 3, second row)

additional uncertainty is introduced by the metric practitioner when defining the time-dependency of the $\Delta\alpha$ perturbation, which may be contrasted to IPCC emission metrics where the temporal evolution of the perturbation (i.e., atmospheric concentration change) is pre-defined (or rather, lifetimes and decay functions of the various forcing agents). Likewise, the *RF* models employed to give radiative efficiencies for various forcing agents are pre-defined by the IPCC -- models having origins linked to standardized experiments employing rigorously evaluated radiative transfer and/or climate models, which may be

contrasted to the models applied to estimate $RF_{\Delta\alpha}$ which can vary widely in their complexity and uncertainty (for a brief review of these, see Bright & O'Halloran (2019)).

## 4 Quantitative metric evaluation

The metrics presented in Section 3 are systematically compared quantitatively henceforth by deriving them for a set of common cases, starting first with the metrics applied to yield a time series of $CO_2$-eq. pulse emissions (or removals) in time. For all
calculations, the assumed climate "efficacy" (Hansen et al., 2005) – or the global climate sensitivity of $RF_{\Delta\alpha}$ relative to $RF_{CO_2}$ – is 1.

### 4.1 CO₂-eq. pulse time series measures

Let us first consider a geoengineering case where one m² of a rooftop is painted white during the first year of a 100-year simulation which increases the annual mean surface albedo (Fig. 3 A) for the full simulation period resulting in a constant
negative $RF_{\Delta\alpha}$ (Fig. 3 B). The objective is to estimate a series of $CO_2$-eq. fluxes associated with the local $RF_{\Delta\alpha}(t)$.

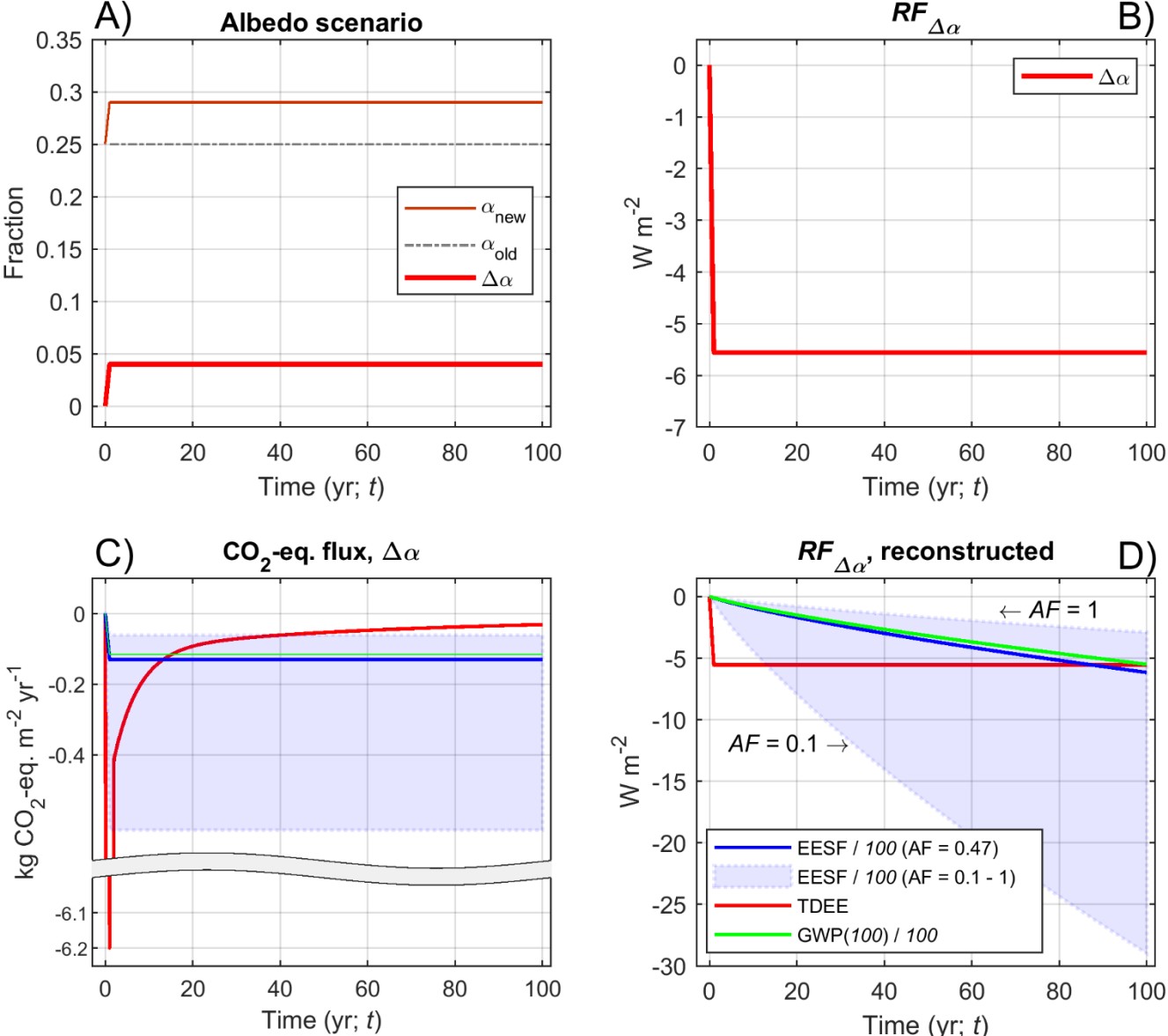

**Figure 3. Example application of metrics yielding a complete time series of $CO_2$-eq. pulse emissions or removals. A)** Time-dependent local $\Delta\alpha$ scenario ("$\Delta\alpha$" = $\alpha_{new}$ - $\alpha_{old}$); **B)** The corresponding local annual mean instantaneous shortwave radiative forcing over time ($RF_{\Delta\alpha}(t)$); **C)** The derived metrics *TDEE*, *GWP(100)/100*, and *EESF/100* for a range of airborne fractions (AF), ; **D)** The reconstructed local annual mean $RF_{\Delta\alpha}(t)$ based on the values shown in panel C) and Eq. **(4)**. Note that the legend in panel D) also applies to panel C).

Figure 3 C presents the results after applying the relevant metrics to the common $RF_{\Delta\alpha}$ and time-dependent $\Delta\alpha$ scenario. To assess their fidelity or "accuracy", the resulting $CO_2$-eq. series of annual $CO_2$ pulses (in this case removals) are used with Eq. (4) to re-construct the $RF_{\Delta\alpha}$ time profile (Fig. 3 B). Unsurprisingly, annual $CO_2$-eq. removals estimated with the *TDEE*

approach (Fig. 3 C) exactly reproduce $RF_{\Delta\alpha}$, and thus the two red curves shown in Figures 2B and D are identical (note the difference in scale).   Figure 3 C illustrates the sensitivity of the *EESF*-based measure derived using an AF of 0.47 (mean of the last seven years based on the most recent global carbon budget, e.g. Friedlingstein et al., (2019); Fig. 1) relative to a broad range of AF values (note that the result obtained using AF = 1 is referred to as the "time-independent emissions equivalent (*TIEE*)" presented in Bright et al. (2016)).   Irrespective of the AF value that is chosen, when applied in a forward-looking analysis utilizing a time-dependent $\Delta\alpha$ scenario with a time horizon of 100-yrs., the *EESF* approach underestimates the magnitude of the annual $CO_2$-eq. pulse occurring in the short-term relative to *TDEE* (Fig. 3 C) and hence also $RF_{\Delta\alpha}$ in the short-term (Fig. 3 B & D).   This is because the $CO_2$ forcing represented as $TH^{-1}k_{CO2}AF$ with the *EESF* approach is weaker than the $CO_2$ forcing represented as $k_{CO2}\sum_{t=0}^{t=TH} y_{CO_2}(t)$ with the *TDEE* approach in the short-term.   For higher AF values, annual $CO_2$-eq. removals estimated using the *EESF*-based approach will underestimate the $RF_{\Delta\alpha}$ at each time step (Fig. 3 D), despite the higher magnitude $CO_2$-eq. estimate (relative to *TDEE*) seen in the longer-term (Fig. 3 C).   This is owed to the lower atmospheric $CO_2$-equivalent abundance that is accumulated over the period when the series of annual $CO_2$-eq. fluxes are reduced to compensate for the higher *AF*.

For *TH* = 100 years, the *EESF*-based estimate will always be lower in magnitude in the short-term and higher in magnitude in the longer-term relative to *TDEE* (Fig. 3 C).   The same is also true for the annual *GWP*-based $CO_2$-eq. estimate, although at least the reconstructed $RF_{\Delta\alpha}$ value at $t = TH$ will always be identical to the actual $RF_{\Delta\alpha}$ value at $t = TH$ (Fig. 3 D).   In general, *EESF*- and *GWP*-based estimates of annualized $CO_2$-eq. emissions (or removals) are sensitive to the chosen *TH* and will always exceed (in magnitude) estimates based on *TDEE*.   This is demonstrated in Figure 4.

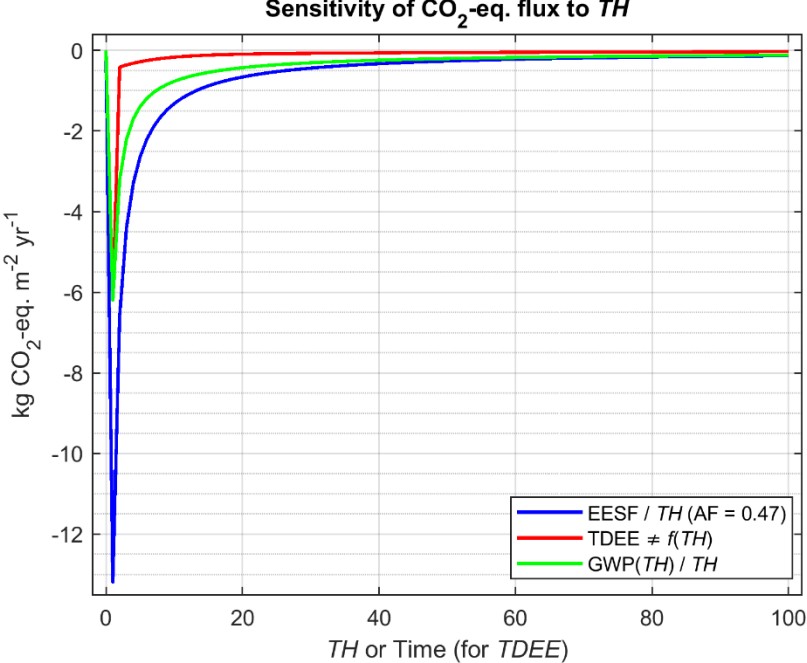

**Figure 4. Magnitude of the annual CO2-eq. emission (removal) pulse as a function of metric *TH* for the *EESF* and *GWP* measures relative to *TDEE* which is insensitive to *TH*.**

The *EESF*-based estimate in this example is higher (in magnitude) than the GWP-based estimate because the assumed AF of 0.47 is lower than the mean atmospheric fraction following pulse emissions (i.e., $y_{CO_2}(t)$) over the range of time horizons shown (the mean atmospheric fraction at $TH = 100$ when applying Joos et al. (2013) function is 0.53). In contrast to the *EESF*- and *GWP*-based approaches, the magnitude of the annual CO$_2$-eq. removals estimated with *TDEE* is insensitive to the chosen *TH*.

## 4.2 Single CO$_2$-eq. pulse measures

Turning our attention now to measures yielding a single CO$_2$-eq. emission or removal pulse, let us now consider a forest management case where managers are considering harvesting a deciduous broadleaf forest to plant a more productive evergreen needleleaved tree species. It is known that when the evergreen needleleaf forest matures in 80-years its mean annual surface albedo will be about 2% lower than the deciduous broadleaved forest. The corresponding annual local $RF_{\Delta\alpha}$ at year 80 is 1.8 W m$^{-2}$, and we wish to associate a CO$_2$-equivalence to this value in order to weigh it against an estimate of the total CO$_2$ stock difference between the two forests after 80 years (i.e., $TH = 80$). Assuming we have no information about how the albedo evolves *a priori* in the two forests before year 80, we have no choice but to apply the *EESF* measure.

Figure 5 presents the $CO_2$-eq. estimate based on *EESF* for an AF range of $0.1 - 1$, shown together with an estimate in which the AF is obtained using the mean fraction of $CO_2$ remaining in the atmosphere at 80 years following an emission pulse, obtained from the latest IPCC impulse-response function ($y_{CO_2}(t)$), and with the highest and lowest airborne fractions of the last 7 years.

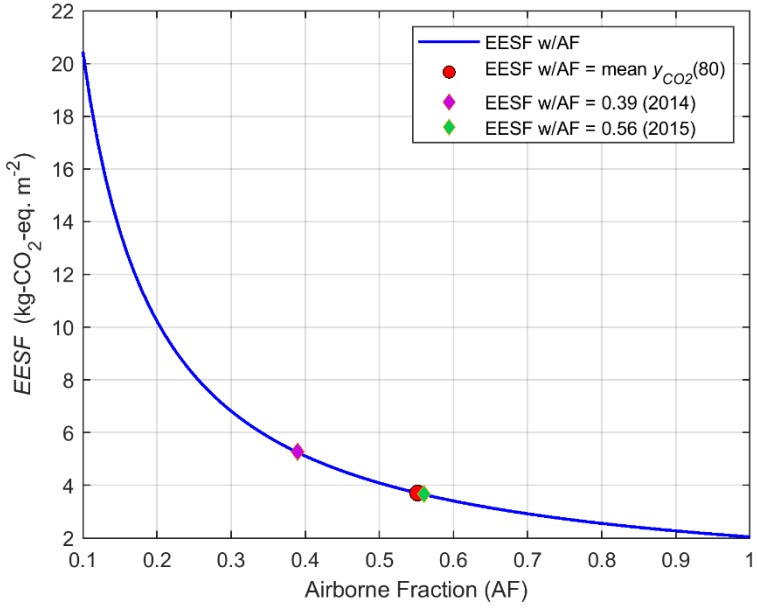

**Figure 5.  Sensitivity of *EESF* to the airborne fraction (AF).**

Figure 5 illustrates *EESF*'s sensitivity to the assumed AF.  For instance, *EESF* with AF = 0.3 is double that estimated with AF = 0.6 – a normal AF range for the past 60 years (Fig. 1).  *EESF* estimated using AF from 2015 (Fig. 5, green diamond) is 44% lower than *EESF* using AF from the previous year (Fig. 5, magenta diamond).  If surface albedo is ever to be included in forestry decision-making – as some have proposed (Thompson et al., 2009a;Lutz and Howarth, 2014) – the subjective choice of the AF becomes problematic given this large sensitivity.  For instance, if the decision-making basis in this example depends on the net of the $CO_2$-eq. of $\Delta\alpha$ and a difference in forest $CO_2$ stock of 4.5 kg $CO_2$ m$^{-2}$, adopting an AF of 0.5 might lead to a decision to plant the new tree species given that the stock difference would exceed the *EESF* estimate (i.e., $CO_2$ sinks dominate), whereas adopting an AF of 0.4 might lead to a decision to forego the planting given that the $CO_2$-eq. of $\Delta\alpha$ would exceed the stock difference (i.e., surface albedo dominates).

Now let's assume the metric user *does* have insight into how the surface albedos of both forest types will evolve over the full rotation period.  In this new example, harvesting the deciduous broadleaf forest to plant an evergreen needleleaf species will first increase the surface albedo in the short-term, yet as the evergreen needleleaf forest grows and tree canopies begin to close and mask the surface, the albedo difference ($\Delta\alpha$) reverts to negative and stays negative for the remainder of the rotation.  This

results in an annual mean local $RF_{\Delta\alpha}(t)$ profile that is first negative and then positive, which is depicted in Figure 6 A (blue solid curve, left y-axis).

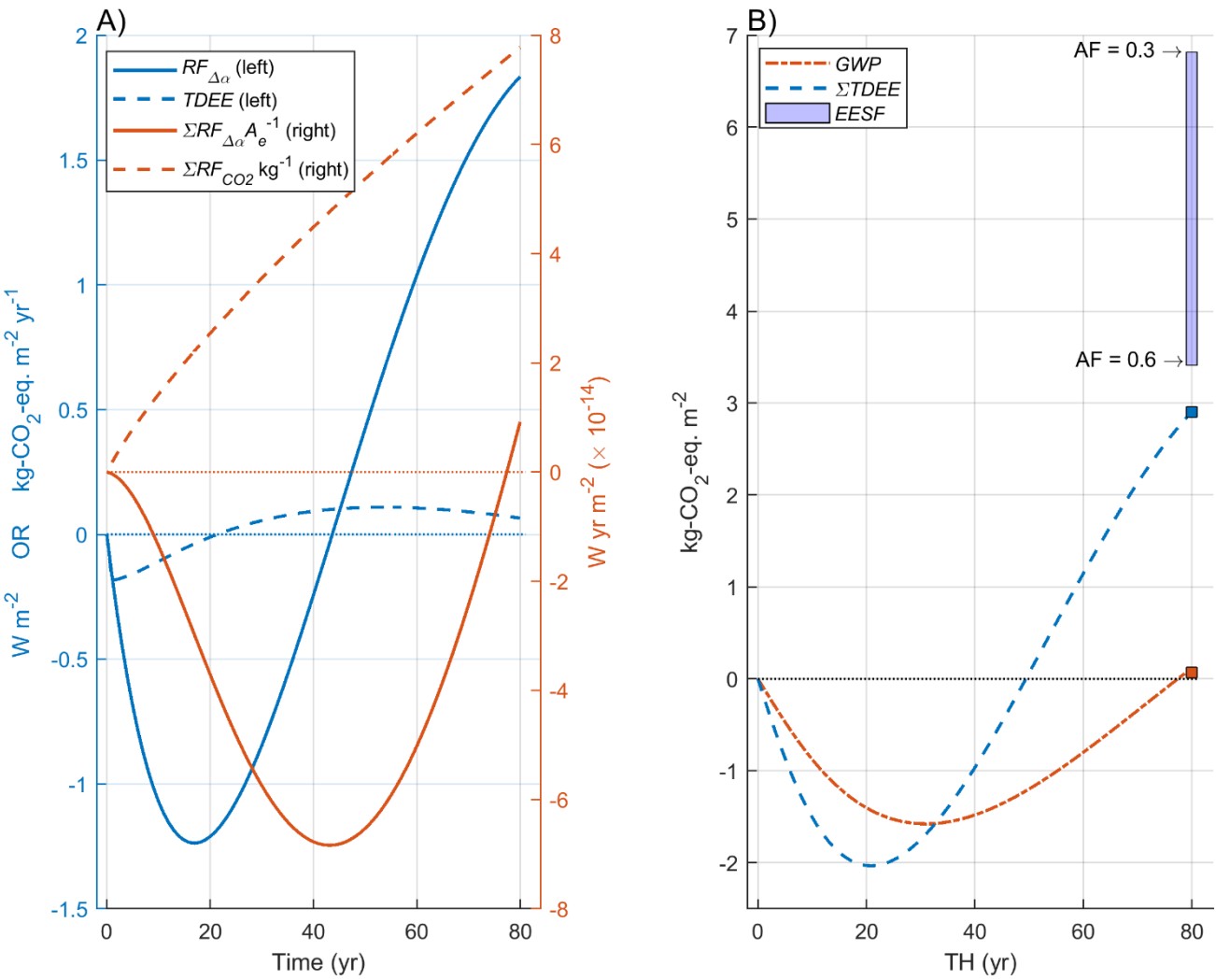


**Figure 6. Example application of metrics yielding a single CO₂-eq. emission (or removal) pulse following a hypothetical forest tree species conversion.** A) $RF_{\Delta\alpha}(t)$ and corresponding $TDEE$ (left y-axis, blue curves) and the temporally-accumulated $RF_{\Delta\alpha}(t)$ normalized to Earth's surface area (solid red, right y-axis) and temporally-accumulated $RF_{CO_2}(t)$ (dashed red, right y-axis) following a 1 kg pulse emission ; B) $EESF$ estimated for the $\Delta\alpha$ (and $RF_{\Delta\alpha}$) occurring at $TH = 80$ shown in relation to $GWP(TH)$ – or the ratio of two red curves shown in panel A – and $\sum TDEE$ estimated at all $THs$.


Converting the $RF_{\Delta\alpha}(t)$ time profile first to a time series of CO₂-eq. emission/removal pulses (i.e., *TDEE*, Fig. 6 A, dashed blue curve) then summing to year 80 gives a measure of the total quantity of CO₂-eq. emitted (or removed) at year 80 – or $\sum TDEE$ (Fig. 6 B, blue curve). $\sum TDEE$ thus "remembers" the negative $\Delta\alpha$ in the early phases of the rotation period (short-

term), leading to a lower CO₂-eq. estimate at year 80 relative to *EESF* estimates computed with airborne fractions of 0.66 and

lower. Similarly, the *GWP*-based estimate "remembers" the negative $\Delta\alpha$ occurring in the short-term; however, *GWP* is a normalized measure, meaning that the time-evolving radiative effects of $\Delta\alpha$ and $CO_2$ are first computed independently from each other prior to the $CO_2$-equivalence calculation, whereas for *TDEE* (and hence $\sum TDEE$) $CO_2$-equivalence depends *directly* on the time-evolving radiative effect of $\Delta\alpha$. Framed differently, $\sum TDEE$ remembers prior $CO_2$-eq. fluxes yielding the

radiatively equivalent effect of the time-dependent $\Delta\alpha$ scenario, whereas the "memories" of $RF_{\Delta\alpha}$ and $RF_{CO2}$ underlying the *GWP*-based $CO_2$-equivalent estimate are first considered in isolation (Fig. 6 A, red curves). Hence the *GWP*-based $CO_2$-eq. estimate in this example is much lower than the $\sum TDEE$-based estimate since the temporally-accumulated $RF_{CO2}$ following a unit pulse emission at $t = 0$ (or $\Sigma RF_{CO2}$, also known as the Absolute GWP or $AGWP_{CO2}$, Fig. 6A dashed red curve) is significantly larger than the temporally-accumulated $RF_{\Delta\alpha}$ (or $\Sigma RF_{\Delta\alpha}$) representing brief periods of both positive- and negative

$RF_{\Delta\alpha}$. Comparing brief or "short-lived" *RF*s with $CO_2$ *RF*s using *GWP* has been heavily criticized for reasons we discuss further in Section 6.

When scalar metrics are required, Figure 6 illustrates the large inherent risk of applying a static measure like *EESF* to characterize $\Delta\alpha$ in dynamic systems. Moreover, for dynamic systems in which $\Delta\alpha$'s time-dependency is defined *a priori*, Fig. 6 illustrates the importance of clearly defining the time horizon at which the physical effects of $\Delta\alpha$ and $CO_2$ are to be compared:

*GWP* gives an effect measured in terms of a present-day $CO_2$ emission (or removal) pulse, while $\sum TDEE$ gives an effect measured in terms of a future $CO_2$ emission (or removal). In other words, internal consistency between the ecological and metric time horizons is relaxed with *GWP* but preserved with $\sum TDEE$.

## 5 Qualitative metric evaluation

The reviewed metrics and underlying methods for converting shortwave radiative forcings from $\Delta\alpha$ (i.e., $RF_{\Delta\alpha}$) into their $CO_2$

equivalent effects – summarized in Table 4 – can primarily be differentiated by the physical interpretation of the derived measure and by whether or not a time-dependency (inter-annual) for $\Delta\alpha$ was defined *a priori*.

**Table 4.** **Overview of distinguishing attributes, methodological differences, drawbacks, and merits of the six $\Delta\alpha$ metrics applied in the scientific literature included in this review.**

| $\Delta\alpha$ Metric | $CO_2$-equivalence Interpretation | Time-dependent $\Delta\alpha$ scenario | Drawbacks | | Merits |
|---|---|---|---|---|---|
| *EESF* | Single pulse | No | - Sensitive to choice of airborne fraction (AF) <br> - Not forward-looking <br> - No carbon cycle dynamics | - | Easy to apply; No need to define a $\Delta\alpha$ scenario *a priori* |

| | | | | |
|---|---|---|---|---|
| *EESF/TH* | Series of uniform pulses | No | - Same as above<br>- $CO_2$-eq. series does not reproduce $RF_{\Delta\alpha}(t)$ [a]<br>- Sensitive to *TH* | - Easy to apply |
| *TDEE* | Series of non-uniform pulses | Yes | - Not scalar | - $CO_2$-eq. series reproduces $RF_{\Delta\alpha}(t)$<br>- Can be compared to an emission scenario<br>- Insensitive to *TH* |
| *ΣTDEE* | Accumulation of a series of non-uniform pulses | Yes | - Cannot be compared to a $CO_2$ pulse of the present | - Compatible with policy targets based on cumulative emissions<br>- Insensitive to *TH* |
| *GWP* | Single pulse | Yes | - Sensitive to *TH*<br>- May be a poor indicator of impact when $\Delta\alpha(t)$ is shorter than *TH*. | - Well-known; IPCC conformity<br>- Compatible with IPCC assessments and UNFCCC accounting conventions |
| *GWP(TH) /TH* | Series of uniform pulses | Yes | - Sensitive to *TH*<br>- $CO_2$-eq. series does not reproduce $RF_{\Delta\alpha}(t)$ except at $t = TH$ | - *GWP* method is well-known |

[a] The exception is at $t = TH$ when $AF = TH^{-1} \int_{t=0}^{t=TH} y_{CO_2}(t)\,dt$

For cases when $\Delta\alpha$'s time-dependency is not known or defined *a priori*, the *EESF* measure is the only applicable measure of those reviewed, although it was shown here to be highly sensitive to the value chosen to represent $CO_2$'s airborne fraction (AF; Fig. 5) – a key input variable taking on a wide range of values depending on how it was defined.  In general, when AF is
defined according to historical accounts of global carbon cycling, its value is prone to large fluctuations across short time scales (Fig. 1) due to natural variability in the global carbon cycle (Ciais et al., 2013).  When defined as the fraction of $CO_2$ remaining in the atmosphere following a pulse emission – as would be obtained from a simple carbon cycle model (i.e., a $CO_2$ impulse-response function) -- its value depends on the time horizon chosen and underlying model representation of atmospheric removal processes (i.e., time-constants).  Use of the latter definition of AF affixes a forward-looking time-
dependency to the *EESF* measure which is inconsistent with the definition of $\Delta\alpha$ and adds subjectivity (i.e., the choice in *TH*). Basing the AF on global carbon budget reconstructions would at least preserve some element of objectivity, although given

the measure's sensitivity to AF it would be prudent to compute the measure for a range of AFs (i.e., as constrained by the observational record) in effort to boost transparency. Forgoing the use of an AF altogether would eliminate all subjectivity, as has been suggested elsewhere (Bright et al., 2016).


For cases involving a time-dependent $\Delta\alpha$ scenario that is defined *a priori*, forward-looking measures are identified whose methodological differences give rise to different interpretations of "CO₂-equivalence" (Table 4). For example, the *GWP* measure can be interpreted as CO₂-eq. pulse emitted at present yielding the accumulated radiative forcing of the $\Delta\alpha$ scenario at *TH* years into the future. *GWP* has merit from the standpoint that it is easy to apply and conforms to established reporting methods, accounting standards, or decision-support tools such as Life Cycle Assessment (e.g., Cherubini et al. (2012); Sieber et al. (2020)). Scientifically, however, there are important limitations to *GWP* when the forcing (i.e., $\Delta\alpha$) is short-lived or temporary (Allen et al., 2016;Pierrehumbert, 2014;Allen et al., 2018;Lynch et al., 2020;Cain et al., 2019). The *TDEE* measure, on the other hand, can be interpreted as a complete time series of CO₂ emission pulses (i.e., a complete emission scenario) yielding the instantaneous radiative forcing of the $\Delta\alpha$ scenario. When summed to *TH*, the latter (as *ΣTDEE*) provides a clearer indication of the radiative impact incurred up to *TH*, thus having greater scientific merit as an indicator of future warming.

The permutations of *GWP* and *EESF* applied to arrive at a time series of CO₂-eq. pulses -- *GWP(TH)/TH* and *EESF/TH* -- have little merit on the grounds that the resulting series does not reproduce $RF_{\Delta\alpha}(t)$ (Fig. 3 D). The *TDEE* approach was proposed to overcome this limitation, although it should be stressed that – like *GWP(TH)/TH* – its derivation requires that a time-dependent $\Delta\alpha$ scenario be defined *a priori*, which adds uncertainty and may not always be possible.

## 6 *GWP\** and $\Delta\alpha$

It has been known that the conventional usage of *GWP* does not adequately capture different behaviors of short-and long-lived climate pollutants or their impact on global mean surface temperatures (Pierrehumbert, 2014;Allen et al., 2016;Shine et al., 2003;Fuglestvedt et al., 2010). Some have proposed an alternative usage of *GWP* – denoted *GWP\** (Allen et al., 2018) -- which overcomes this problem by equating an increase in the emission rate of a short-lived climate pollutant (or radiative forcing agent) with a one-off "pulse" CO₂ emission. *GWP\** recognizes that a pulse emission of CO₂ and a sudden step-change in the sustained rate of emission of a short-lived climate pollutant (SLCP) both give near-constant radiative forcing. Or, alternately, that a progressive linear increase (or decrease) in the rate of an SLCP emission is approximately equivalent to a sustained step change in the emission rate of CO₂. As such, *GWP\** is considered to have greater "environmental integrity" than the conventional *GWP* metric (Allen et al., 2018), as it is better fit to serve the purpose of a measure of progress towards a global temperature-oriented climate goal (i.e., limit warming to "well below 2°C"). Compared to conventional *GWP*, cumulative CO₂-eq. emissions based on *GWP\** provide a clearer indication of future warming, and future CO₂-eq. emission rates better-indicate future warming rates. *GWP\** thus better-relates all climate pollutants in a common cumulative emission

(or emission budget) framework, making it easier to formulate mitigation strategies that provide a more accurate indication of
progress towards climate stabilization.

Among one of the more distinguishing features of *GWP\** is that, when applied to radiative forcings rather than pulse emissions,
information about the time-dependency of the perturbation (i.e., the lifetimes of "climate pollutants" or forcing agents) is not
required (Lee et al., 2021;Cain et al., 2019;Allen et al., 2018), making it an attractive alternative to *EESF*. In other words, a
*GWP* estimate of the "short-lived" forcing agent under scope – which requires such information to be known or defined *a
priori* -- is unnecessary in its calculation. Only the rate of change of the forcing is required, scaled by *TH/AGWP(TH)$_{CO2}$* as
follows (Lee et al., 2021;Allen et al., 2018):

$$E_{CO2-eq.*} = \frac{TH}{AGWP(TH)_{CO2}} \left( \frac{\Delta RF_{\Delta\alpha}}{\Delta t} \right)$$  (9)


where *TH* is the time horizon, *AGWP(TH)$_{CO2}$* is $CO_2$'s AGWP at the same *TH* (i.e., $9.2 \times 10^{-14}$ W m$^{-2}$ kg$^{-1}$ year when *TH* = 100
yrs.), $\Delta t$ is the time step change, and $\Delta RF_{\Delta\alpha}$ is the time-differential of $RF_{\Delta\alpha}(t)$ over the step change. $E_{CO2-eq.*}$ thus represents
the $CO_2$-eq. emission pulse for the step change and will equal *EESF* when the AF (in Eq. 6 denominator) corresponds to the
mean of $y_{CO_2}(t)$ over the *TH* (i.e., $TH^{-1} \int_{t=0}^{t=TH} y_{CO_2}(t)\, dt$). A *TH* of 100 years is typically applied in Eq. (9) which is justified
when it exceeds the lifetime of the SLCP or when the time-integrated radiative forcing of the forcing agent (i.e, $\Delta\alpha$) becomes
a constant at this time scale, since the time-integrated radiative forcing of the reference gas (i.e., *AGWP$_{CO2}$*) increases linearly
with *TH*. In other words, the *TH*-dependence cancels out in the calculation of $CO_2$-eq.\*, rendering *GWP\** insensitive to the
choice in *TH* which contrasts with the conventional *GWP* (Allen et al., 2016; Allen et al., 2018). The step change $\Delta t$ for which
$\Delta RF$ is calculated is typically taken as 20 years to "reduce the volatility of $CO_2$-eq.\* emissions in response to variations in
SLCP emission rates" (Allen et al. 2018; Cain et al. 2019), although comprehensive investigations into the appropriateness of
this choice when applied to a wide variety of time-varying SLCP emission (radiative forcing) scenarios are lacking. We note
that more recent works (Cain et al., 2019; Lee et al., 2020) employed weighting-based modifications to Eq. (9) in effort to
better-account for the longer-term temperature equilibration to past forcing changes:

$$E_{CO2-eq.*} = \left[ (1-s) \frac{TH}{AGWP(TH)_{CO2}} \right] \frac{\Delta RF_{\Delta\alpha}}{\Delta t} + s \frac{\overline{RF_{\Delta\alpha}}}{AGWP(TH)_{CO2}}$$  (10)

where *s* is a factor weighting the delayed response by global mean temperature to the radiative forcing history, represented
here (following Lee et al., 2020) as the mean forcing over the period $\Delta t$ – or $\overline{RF_{\Delta\alpha}}$. Note that *s* is analogous to the "$\alpha$" term
seen in Eq. (1) of Lee et al. (2020) and that the factor 1-*s* is analogous to the rate contribution weight denoted as "*r*" in Eq. S1
of Cain et al. (2019). Like the choice of $\Delta t$, however, few investigations have been carried out to assess the appropriateness

of weight sizes applied in Eq. (10) for different SLCP emission (radiative forcing) scenarios having widely-varying temporal dynamics.

We explore the sensitivity of the choice in both $\Delta t$ and $s$ on $CO_2$-eq. emissions (removals) estimated with the modified *GWP\** approach (Eq. (10)) for three hypothetical local $RF_{\Delta\alpha}(t)$ scenarios presented in Figure 7 A, the first of which ("Scenario A") being identical to the forest management scenario plotted in Figure 6 A and extended by 20-yrs, which is characterized by a negative *RF* in short-term and positive *RF* in the longer-term (Fig. 7 A, blue). $RF_{\Delta\alpha}(t)$ in Scenario B corresponds to a linearly increasing $\Delta\alpha$ trend which is loosely analogous to incremental deforestation occurring on a regional scale (Fig. 7 A, red), and Scenario C resembles a permanent albedo decrease, analogous to urban expansion into a cropland (Fig. 7 A, yellow).

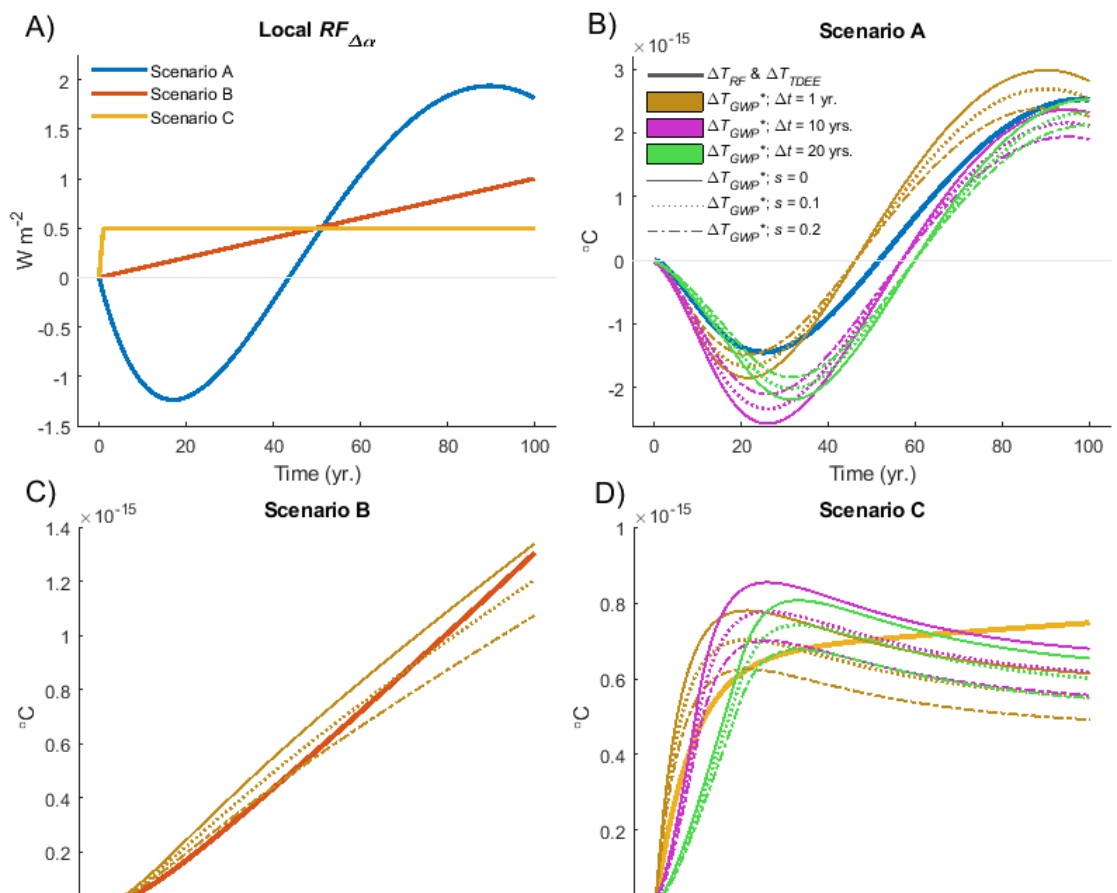

Figure 7. Performance of *GWP\** computed for three stylized scenarios of surface albedo change-driven radiative forcing using Eq. (10) with nine different parameter sets. A) Local radiative forcing of one permanent and two temporally-evolving surface albedo

 **change scenarios; B-D) The corresponding global mean temperature response $\Delta T$ to the radiative forcing relative to that which has been reconstructed using the $CO_2$-eq. emission (removal) time series computed with *TDEE* and *GWP\** under the assumption that $\Delta \alpha_{t+n-t}$ is known. $\Delta T$ in panels B-D is estimated with a temperature impulse response function following (Boucher and Reddy, 2008;Myhre et al., 2013) having a climate sensitivity of 1.06 K (W m$^{-2}$)$^{-1}$, which is equivalent to a 3.9 K equilibrium climate response to an abrupt $CO_2$ concentration doubling.**


We then reconstruct the global mean temperature response ($\Delta T$) of the $CO_2$-eq.\* emission (removal) scenario under varying assumptions surrounding the size of $\Delta t$ and the weighting factor *s* (shown in Fig. 7 B legend), which is then compared to the $RF_{\Delta \alpha}$-based $\Delta T$ and the $\Delta T$ reconstructed using the $CO_2$-eq. emission (removal) scenario based on the *TDEE* approach (Fig. 7 B-D). For Scenario A (Fig. 7 B), we find no obvious parameter set that outperforms any other in terms of the faithfulness by which the $CO_2$-eq\* emission (removal) scenario reproduces $\Delta T$ across the full time horizon. There appears to be a trade-off between the near- and long-term reproduction accuracy of different parameter sets: a 20-yr $\Delta t$ with no weighting (Fig. 7 B, solid green curve) better-reproduces the $\Delta T$ response seen in the short-term (~ <20 yrs.) as well as the $\Delta T$ seen at the end of the scenario time horizon (year 100), whereas a 10-yr. $\Delta t$ with no weighting (Fig. 7 B, solid purple curve) better-reproduces the $\Delta T$ response seen in the longer-term (from ~60-90 yrs.). An increase to the weighting factor *s* serves to dampen the amplitude between the maximum cooling and warming seen in the short- and longer-terms, respectively (Fig. 7 B, spread between like-colored curves). As for Scenario B representing a linear increase in *RF*, the reconstructed $\Delta T$ is insensitive to $\Delta t$ and thus only results for a 1-year $\Delta t$ are computed and presented in Figure 7 C. Although a weighting factor of 0.2 is most accurate for the first ~50 years, a weight of 0.1 gives a more faithful $\Delta T$ reproduction for the full time period. As for Scenario C representing a step change in *RF* (Fig. 7 D), again we find no obvious parameter set that yields a faithful $\Delta T$ reproduction across the full time period. High *s* weights overpredict $\Delta T$ in the medium-term but reproduce $\Delta T$ best in the longer-term (Fig. 7 D, solid curves), while a $\Delta t$ larger than 10 years appears to result in large underpredictions in the short-term (i.e., < ~ 20 yrs; Fig. 7 D, green curves).

Unsurprisingly, $\Delta T$ reconstructed using the $CO_2$-eq. emission (removal) scenario estimated with the *TDEE* approach exactly reproduces the *RF*-based $\Delta T$ and thus these two estimates are plotted jointly as a single curve in Figure 7 B-D (wider solid curves). Thus, when future surface albedo changes are defined *a priori* (i.e., when the $\Delta \alpha$ perturbation "lifetime" is known or estimated), a $CO_2$-eq. emission (removal) time series quantified with *TDEE* is far superior to one based on *GWP\** irrespective of the choice in $\Delta t$ or weight sizes applied, making it the better $CO_2$-eq. measure of progress towards global temperature stabilization.

## 7 Spatial disparity in climate response between $CO_2$ emissions and $\Delta\alpha$ perturbations

The climate (i.e., temperature) response to a $\Delta\alpha$ perturbation either isolated (e.g., Jacobson and Ten Hoeve, (2012)) or as part of LULCC (e.g., (Pongratz et al., 2010;Betts, 2001)) is highly heterogeneous in space, the magnitude and extent of which depends on its location (Brovkin et al., 2013;de Noblet-Ducoudré et al., 2012). This is because the response pattern of climate feedbacks has a strong spatial dependency – feedbacks are generally larger at higher latitudes due to higher energy budget sensitivity to clouds, water vapor, and surface albedo, which generally increases the effectiveness of $RF$ in those regions (Shindell et al., 2015). This is in contrast to $CO_2$ emissions where both $RF$ and the temperature response are more homogeneous in space (Hansen and Nazarenko, 2004;Hansen et al., 2005;Myhre et al., 2013). This has caused some researchers to question the utility of a CO2-eq. measure for $\Delta\alpha$ (Jones et al., 2013) or encouraged others to look for solutions or further methodological refinements. For instance, some researchers (e.g., Cherubini et al. (2012); Zhao & Jackson (2014)) have applied climate "efficacies" – or the climate sensitivity of a forcing agent relative to CO2 (Joshi et al., 2003;Hansen et al., 2005) – to adjust $RF_{\Delta\alpha}$ prior to the CO2-eq. calculation. Such adjustments recognize that the temperature response to $RF$ depends on the geographic location, extent, and type of underlying forcing associated with the $\Delta\alpha$ (e.g., land use/land cover change (LULCC), white-roofing, etc.) which can be co-associated with other perturbations (Table 5) like those arising from changes to vegetative physical properties (for the LULCC case) which can modify the partitioning of turbulent heat fluxes above and beyond the purely radiatively-driven change (Davin et al., 2007;Bright et al., 2017).

Using a climate efficacy to adjust $RF_{\Delta\alpha}$, however, is not without its drawbacks. A first and obvious drawback is that efficacies are climate model dependent (Hansen et al., 2005;Smith et al., 2020;Richardson et al., 2019). Climate models vary in their underlying physics, which is evidenced by the large spread in $CO_2$'s climate sensitivity across CMIP6 models (Meehl et al., 2020;Zelinka et al., 2020). A second drawback is that climate sensitivities for certain forcing agents like $\Delta\alpha$ are tied to experiments that differ largely in the way forcings have been imposed in time and space. Both drawbacks contribute to large uncertainties in the choice of efficacy for $\Delta\alpha$. The latter drawback is especially problematic since the $\Delta\alpha$ perturbation is often accompanied by perturbations to other surface properties and fluxes (Table 5) having large spatial- and temporal dependencies. The turbulent heat flux perturbations that accompany a net radiative flux change at the surface affect atmospheric temperature and humidity profiles (Bala et al., 2008;Modak et al., 2016;Schmidt et al., 2012;Kravitz et al., 2013), causing the atmosphere to adjust to a new state resulting in a net radiative flux change at TOA that extends beyond the instantaneous shortwave radiative flux change (i.e., $RF_{\Delta\alpha}$).

**Table 5. Differences in surface property and flux perturbations between geoengineering-type forcings involving non-vegetative solar radiation management (SRM) and forcings from LULCC, land management change (LMC), or forest management change (FMC). "$\Delta r_a$" = change to bulk aerodynamic resistance; "$\Delta r_s$" = change to bulk surface resistance; "$\Delta\lambda(E)$" = latent heat flux change from a change to evaporation; "$\Delta\lambda(E+T)$" = latent heat flux change from a change to both evaporation and transpiration; "$\Delta H$" = sensible heat flux change**

| Forcing type | Surface property perturbation | Surface flux perturbation |
|---|---|---|
| Geoengineering (non-veg. SRM) | $\Delta\alpha$ | $\Delta\lambda(E)$, $\Delta H$ |
| LULCC; LMC; FMC | $\Delta\alpha$, $\Delta r_a$, $\Delta r_s$ | $\Delta\lambda(E+T)$, $\Delta H$ |


For example, the efficacy of LULCC forcing across the six studies reviewed by Bright et al. (2015) ranged from 0.5 to 1.02 owed to differences in model set-up (e.g., fixed SST vs. slab vs. dynamic ocean), differences in the spatial extent and magnitude of the imposed LULCC forcing (e.g., historical transient vs. idealized time slice), as well as the LULCC definition (i.e., the type of LULCC that was included in the study such as only afforestation/deforestation vs. all LULCC). Even when controlling
for differences in experimental design (e.g., CMIP protocols), the climate efficacy of historical LULCC has been found to vary considerably in both sign and magnitude (c.f. Figure 8, Richardson et al. 2019), which is more likely attributed to the larger spread in effective radiative forcing (*ERF*) for LULCC than for $CO_2$. For instance, Smith *et al.* (2020) report a standard deviation of 6% in the *ERF* of $CO_2$ (4×abrupt) across 17 GCMs/ESMs participating in RFMIP in contrast to 175% for LULCC, although it should be kept in mind that the *ERF* is weak for LULCC thus relative differences become large.


An additional drawback and source of uncertainty underlying efficacies is related to differences in their definition. Differences in definition can stem from either different definitions of *RF* itself or differences in the definition of the temperature response per unit *RF* (Richardson et al., 2019;Hansen et al., 2005). Regarding the latter, most base the temperature response for $CO_2$ on the equilibrium climate sensitivity (ECS) for a $CO_2$ doubling, although good arguments have been made for using the
transient climate response (TCR) instead, particularly for short-lived forcing agents (Marvel et al., 2016;Shindell, 2014). The temperature response for the forcing agent of interest is rarely taken as the equilibrium response although there are some exceptions (e.g. "$E_\alpha$" in Richardson *et al.* 2019 which is based on climate feedback parameters obtained from ordinary least square regressions). Efficacies are also sensitive to the definition of *RF* (Richardson et al., 2019;Hansen et al., 2005). For example, the efficacy of sulfate forcing (5×$SO_4$) has recently been shown to vary from 0.94 to 2.97 depending on whether *RF*
is based on the net radiative flux change at TOA from fixed SST experiments or the instantaneous shortwave flux change at tropopause (Richardson et al., 2019).

Ideally, $CO_2$-eq. metrics based on the *RF* concept should be based on an *RF* definition yielding efficacies approaching unity for a broad range of forcing types. Although there is currently no consensus here, strong arguments have been made for *RF*
definitions based on the net radiative flux change at TOA resulting from fixed SST experiments with GCMs/ESMs (i.e., "$F_s$" in Hansen et al. 2005; "$ERF_{SST}$" in Richardson et al. 2019), since such definitions yield efficacies approaching unity for a broad range of forcing types. However, for most $\Delta\alpha$ metric practitioners it is not feasible to quantify atmospheric adjustments and hence the *ERF*. Efficacies compatible with $RF_{\Delta\alpha}$ (instantaneous $\Delta SW$ at TOA) could be the more feasible option for metric calculations, but broad consensus would need to be established first surrounding appropriate efficacy values for different

forcing types associated with the $\Delta\alpha$ perturbation (Table 5). This is especially true for forcings involving changes to the biophysical properties of vegetation – such as LULCC, forestry, etc. – since these are constructs representing a seemingly myriad combination of biophysical perturbations acting on non-radiative controls (i.e., $\Delta r_a$ and $\Delta r_s$) of the surface energy balance. Building consensus for efficacies applicable to geoengineering-type forcings where the only physical property perturbed is the surface albedo (e.g., white roofing, sea ice brightening, etc.) would be less challenging since the confounding

perturbations to $\Delta r_a$ and $\Delta r_s$ and hence to the turbulent heat fluxes are removed. Nevertheless, irrespective of whether broad scientific consensus can be reached surrounding efficacies suitable for $\Delta\alpha$ metrics, additional responsibility would always be imposed on the metric practitioner to ensure that the chosen efficacy aligns with the forcing type underlying the $RF_{\Delta\alpha}$.

## 8 Discussion

### 8.1 Summary of merits

In this review, we quantitatively and qualitatively reviewed metrics (methods) to characterize $RF_{\Delta\alpha}$ in terms of a $CO_2$-equivalent effect. We note that while many metrics exist, none are true "equivalents" to $CO_2$ due to its unique behavior. The climate effects of the calculated $CO_2$-eq. emissions should ideally be the same regardless of the mix of forcing agents – including $\Delta\alpha$. However, different forcing agents have different physical properties, and a metric that establishes equivalence with regard to one effect cannot guarantee equivalence with regard to other effects and over extended time periods. Differences

among the reviewed $\Delta\alpha$ metrics could be attributed to the different ways of dealing with the time-dependency of $RF_{CO2}$, which to a large extent was determined by whether a time-dependency was defined for the $\Delta\alpha$ perturbation. When the $\Delta\alpha$ perturbation was assumed to have no time-dependency, as was the case for the *EESF* metric, uncertainties arose from the choice of AF giving a mere snapshot in time of the $CO_2$ perturbation. For metrics like *GWP* and *TDEE* that explicitly account for the time-dependency of $RF_{CO2}$, the need to define a time-dependency for $\Delta\alpha$ *a priori* introduces uncertainty owed to the reversible

nature of $\Delta\alpha$. Unlike most climate pollutants having standardized perturbation lifetimes determined by the physics of the earth system, the perturbation "lifetime" of $\Delta\alpha$ is tied to a parcel of land and dictated by future anthropogenic activities occurring on that land. Users should strive to be aware of the limitations and caveats of the reviewed $\Delta\alpha$ metrics - defining a $\Delta\alpha$ time-dependency might improve the precision of the $CO_2$-eq. estimate but not necessarily its accuracy if the future (historical) $\Delta\alpha$ cannot be confidently projected (re-constructed). For instance, application of *EESF* to $\Delta\alpha$ perturbations in dynamic systems

(i.e., systems in which $\Delta\alpha$ exhibits large variation over shorter time scales) opens up the risk for grossly mis-characterizing the system, particularly when the chosen $\Delta\alpha$ is not representative of the mean $\Delta\alpha$ of the system under scope (e.g., Fig. 6 B).

   Although not applied as a $\Delta\alpha$ metric in the studies we included in our review, our review of *GWP\** (Section 6) suggests that it is inferior to *TDEE* as an indicator of future warming when the future time-dependency or "lifetime" of $\Delta\alpha$ is known or defined

*a priori* (Fig. 7 B). However, for cases when the future $\Delta\alpha$ is unknown or deemed too uncertain, one could argue that – as a

scalar metric – *GWP\** has greater scientific merit than *EESF* when applied to step changes in $RF_{\Delta\alpha}$ from the standpoint that $CO_2$'s atmospheric time-dependency is taken explicitly into account. *GWP* – also a scalar metric – has some merit from the standpoint that it is well-known, although scientifically its merits fade when the forcing agent is short-lived (Allen et al., 2018;Lee et al., 2021;Lynch et al., 2020) – as is often the case for *Δα*. As a scalar metric that accounts for *Δα*'s time-dependency, we deem *ΣTDEE* to have greater scientific merit than *GWP* because it is a better indicator of future warming, which is supported quantitatively by the $\Delta T$ reconstructions highlighted in Table 6, based on the $RF_{\Delta\alpha}(t)$ scenarios presented in Figure 7 A.

**Table 6. Comparison of future $\Delta T$ (global mean) from the $RF_{\Delta\alpha}(t)$ scenarios shown in Figure 7 A reconstructed using *GWP*100 and $\sum_0^{100} TDEE$.**

| | Actual $\Delta T$ at $TH = 100$ yrs., °C | Reconstructed $\Delta T$ at $TH = 100$ yrs. using *GWP*100, °C (% of actual) | Reconstructed $\Delta T$ at $TH = 100$ using $\sum_0^{100} TDEE$, °C (% of actual) |
|---|---|---|---|
| Scenario A | $2.52 \times 10^{-15}$ | $5.25 \times 10^{-16}$ (21) | $2.03 \times 10^{-15}$ (80) |
| Scenario B | $-1.30 \times 10^{-15}$ | $-6.22 \times 10^{-16}$ (48) | $-1.19 \times 10^{-15}$ (91) |
| Scenario C | $7.47 \times 10^{-16}$ | $6.16 \times 10^{-16}$ (82) | $7.21 \times 10^{-16}$ (97) |

Although this review has provided needed guidance for choosing appropriate *Δα* metrics according to the context in which they have merit, users should always be mindful that $RF_{CO2}$ and $RF_{\Delta\alpha}$ are not necessarily additive. The global mean temperature may respond differently to identical *RF*s, and, although there are ways to deal with this discrepancy – either by using *ERF*s directly in the metric calculation or by adjusting *RF*s with appropriate efficacy factors. Such approaches require additional modeling tools, which introduces notable additional uncertainties (Section 7). Efficacies for inhomogeneous forcings like $RF_{\Delta\alpha}$ are spatial pattern- and scale-dependent (Shindell et al., 2015), and are sensitive to the climate model set-up and experimental conditions (i.e., how, where, and when *Δα* is imposed in the model). Moreover, efficacies are forcing-type dependent; that is, the forcing signal driving the underlying temperature response may depend on multiple additional perturbations at the surface that are co-associated with *Δα*. A good example is LULCC, which perturbs a suite of additional biogeophysical properties affecting surface fluxes (Table 5), some of which resulting in atmospheric feedbacks (or adjustments) that can counteract the *Δα*-driven signal (Laguë et al., 2019). Since LULCC represents a broad range of land-based forcings, each of which in turn representing a myriad combination of surface biogeophysical property perturbations, the risk of misapplication of efficacies derived from climate modeling simulations of LULCC is inherently large.

## 8.2 Research Roadmap

Research efforts directed towards building a scientific consensus surrounding the most appropriate $RF_{\Delta\alpha}$ estimation method (or model) for use in metric computation would serve to enhance metric transparency and facilitate comparability across

studies. Given the ease and efficiency of applying radiative kernels for $RF_{\Delta\alpha}$ calculations, such efforts might entail systematic evaluations and benchmarking of radiative kernels (e.g., as in Kramer et al. (2019)) for $\Delta\alpha$.


Reducing uncertainty surrounding the efficacy of $RF_{\Delta\alpha}$ associated with a variety of underlying surface forcing types (i.e., specific LULCC conversions, geoengineering methods, etc.) is paramount to reducing the "additivity" uncertainty (Jones et al., 2013) of $RF$-based metrics for $\Delta\alpha$. This can be achieved through extending existing climate modeling experimental protocols (e.g., LUMIP, GeoMIP, RFMIP) or by creating new protocols that seek to systematically quantify the sensitivity of

the global mean temperature response to variations in the spatial pattern, extent, and magnitude of surface and TOA radiative forcings associated with $\Delta\alpha$.

Research is also needed to examine the relevance of accounting for the climate-carbon feedback in $\Delta\alpha$ metrics, given that such feedback is implicitly included in $CO_2$'s impulse-response function (Gasser et al., 2017). Such research should be mindful of

the regional climate response patterns of the various surface forcing types associated with $\Delta\alpha$, and how regional $CO_2$ sinks are affected in turn by the regional response patterns.

Finally, while not a research need *per se*, a discussion between metric scientists and users/policy makers is needed surrounding three topics (Myhre et al., 2013): i) useful applications; ii) comprehensiveness; and iii) the value of simplicity and

transparency. The first involves identifying which application(s) a particular $\Delta\alpha$ metric is meant to serve. We have already shown for instance that the *EESF* metric is not ideal for characterizing dynamic systems. As for comprehensiveness, from a scientific point of view we would ideally wish to be informed about the totality of climate impacts of a $\Delta\alpha$ perturbation at multiple scales (i.e, at both the local and global levels). But a user may often need to aggregate this information, which necessitates trade-offs between impacts at different points in space, between impacts at different points in time, and even

between the choice of metric indicator (e.g., $RF$ vs. $\Delta T$). Related to the value of simplicity and transparency is the question of whether more complex (yet less transparent) model-based metrics (e.g., those based on *ERF*) are valued by users over simple and more transparent metrics based on analytical formulations. The discussion here should weigh their trade-offs: the former may be more cumbersome to apply or more easily misused, whereas the latter may inadequately capture important physical effects or system dynamics.

**8.3 Concluding remarks**

For the past several decades, emission metrics have served useful in enabling users or decision makers to quickly perform calculations of the climate impact of GHG emissions. Their common $CO_2$-equivalent scale has provided flexibility in emissions trading schemes and international climate policy agreements like The Kyoto Protocol. With the advent of the Paris Agreement and a broadened emphasis (Article 4) to include both emissions *and* removals, more attention to land-based

mitigation seems likely, and the need for a way to compare albedo and $CO_2$ on an equivalent scale may increase. This obliges the scientific community to provide users with better tools to do so.

This review has highlighted many of challenges associated with quantifying and interpreting $CO_2$-equvilanent metrics for $\Delta\alpha$ based on the *RF* concept. A variety of metric alternatives exist, each with their own set of merits and uncertainties depending
on the context in which they are applied. The application of metrics always entails user choices, and while some are scientific, others – such as time frame – are policy-related and cannot be informed by science alone. This review has provided guidance to practitioners for choosing a metric with maximum scientific merit and minimum uncertainty according to the specific application context. Going forward, practitioners should always be mindful of the inherent limitations of *RF*-based measures for $\Delta\alpha$, carefully weighing these against the uncertainties of metrics based on impacts further down the cause-effect chain –
such as a change in temperature.

### Acknowledgements

We acknowledge funding from the Research Council of Norway, grant number 254966. We are grateful for comments and feedback provided by Dr. Gunnar Myhre.

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
