# Peer review of "CO2-equivalence metrics for surface albedo change based on the radiative forcing concept: A critical review"

_Atmospheric Chemistry and Physics, 2020_

## Referee Comment (RC1) · Anonymous Referee #1 · 12 Jan 2021

This article provides a review of the literature on climate metrics to evaluate land use change as CO2-equivalent emissions. There are 6 main methods developed over the last 20 years to evaluate the CO2-equivalence of land use changes, of which GWP and sigma-TDEE are the two most recommended by the authors (though both have their own drawbacks). The majority if not all national emissions inventories under the Paris Agreement only take into account CO2-equivalent emissions for a basket of WMGHGs, and clearly there is an omission in the non-GHG forcing. AR5 reviewed metrics for SLCFs in section 8.7 (in any case they are not widely used) and this article highlights the lack of consistency for metrics for surface albedo changes (which are also seldom used in pracitce). In any case, metrics can not yet describe with acceptable accuracy

the resulting climate impact of a particular future socioeconomic scenario (i.e. from an integrated assessment model; see Dension et al. 2019).

**Main comment and recommendation**

I am personally sceptical of the utility of metrics beyond well-mixed GHGs (and even then, with many problems) and I am not convinced that applying CO2-equilvance to land use changes is useful and robust. However, the authors are not necessarily claiming that it is, so the stated purpose of this paper - a review article on land-use metrics - is achieved. Additionally, practitioners in policymaking and life-cycle assessment applications, prone to using metrics without a full appreciation of where the numbers come from, are likely to continue to do so for the foreseeable future, so it is important that methods are critically reviewed and limitations communicated.

I would recommend more discussion as to the limitations of metrics generally and land-use change specifically. Conceptually, the problem for land use is similar to assigning metrics to SLCFs; like land use change, radiative forcing from SLCFs is spatially heterogenous and theoretically (if not practically) reversible over short time scales, and like land use change many SLCFs co-vary with CO2 emissions. Several groups are actively trying to improve metrics, e.g. GWP* (Allen et al 2018) and CGWP and CGTP (Collins et al. 2020). These adjusted metrics have a better link to final climate impacts (in terms of global mean surface temperature changes) and could include the effects of non-GHG forcers (through the CO2-forcing equilvant measure in Allen et al. 2018), though at the risk and expense of introducing additional complexity. Another way in which land use metrics are difficult to apply is that RF from land use change is small and uncertain (the authors do discuss this). Metrics for GHGs work well enough because the emissions to concentrations to radiative forcing (to temperature, in GTP-style metrics) relationships are well-defined, invertible (at least up until Etminan) and known

within acceptable uncertainty.

The alternative to metrics is to use a simple climate model to determine impacts. The two most recommended methods in this article, sigma-TDEE and GWP, need estimates of the radiative forcing of land use change. Therefore, why not cut out the middle man: calculate RF from land use change, calculate or estimate the change in land-use related $CO_2$ sources or sinks for afforestation/deforestation (if that is the intervention of interest, for a geoengineering-style experiment, the land-use $CO_2$ does not need to be perturbed), and run these scenarios in a simple climate model that evaluates the emissions - concentrations - radiative forcing - temperature process, and determine the temperature difference between this and a reference scenario. Several candidate simple climate models are discussed in Nicholls et al. (2020a, 2020b). The required RF estimate can be obtained from the kernel method that the authors describe, or from something like Jones et al. (2015).

**Line-by-line comments**

- 13: equivalent

- 24: Earth (capital E)

- 26: the 13

- 30: "large scale carbon dioxide removal" - mention afforestation specifically here, and why

- 38: "backward looking measure" - explain what is meant here. RF is not necessarily backward looking, and several future projections exist (including for radiative forcing from land use changes)

- 45: UNFCCC

- 47: "debate about GWP as the metric of choice" - perhaps add a citation here. Almost anything from Myles Allen's group 2018-2020 would fit the bill, or Denison et al 2019

- 58: AR4

- 69-70: True up until AR5, where metrics were only done in rigorousness for WMGHGs and there was no evidence to suggest that ERF differed from RF. I would think that ERF would be preferred for metric calculations now, if it differed from RF. Of course we wait for AR6 for this to be defined, though might be worth a mention to future-proof this paper a little.

- 77-85: This paragraph implies why ERF is a better idea than RF; by including tropospheric and land surface adjustments, temperature responses track ERF more closely than RF, and reduces (perhaps elimiates) the need for efficacy scalings.

- 88: is it definitely regression?

- 90: this is outdated: use Etminan et al. 2016 or the re-fitted Etminan relationship in Meinshausen et al. 2020.

- 94-96: the dependency on background state of the reference gas highlights a weakness of metrics generally; more should be made of this.

- 99-101: the Etminan relationship has even modified the RF from 2xCO2 (3.80 W/m2 versus 3.71 in Myhre et al. 1998) but the point here is that a 1ppm change around present day doesn't affect the radiative efficiency of CO2 much, I guess.

- 107-108: g mol-1 is a simpler unit?

- 116: y(t) could be generalised and could represent individual carbon cycle models, as well as the multi-model mean as was used in Myhre et al. 2013

- 121-123: you could explain why this is important for metrics and why metric calculations based on AR5 values might be incorrect under different future assumptions (including for 1.5C targets, where the impulse response function between pre-industrial and present-day states are substantially different as shown by Millar et al. 2017)

- 139: Block and Mauritsen reference should be 2013, not 2015

- 153-154: To use eq. 5 don't you need climate model output? I suppose you could do this based on observations but it won't be globally complete and will still need a fair amount of data crunching. Maybe I've misunderstood something. table 3: Sciusco et al: should it be "on" rather than "no"?

- 177-179: here is another demonstrable weakness of attempting to use a CO2-eq metric for land use change. While AF can be assumed to be = 0.47 in the recent past, there's no guarantee this would hold in the future (Millar et al. (2017), and Jones et al (2013) on AF in RCPs).

- Figure 1: could now be updated to 2019 using the latest GCP update

- 191-192: "approximate time frame..." this half of the sentence doesn't explain what's going on here - does it mean that 47

- 238: is the minus sign meant to be there after TH? Section 3 subsectioning: it's rather top-heavy (most of the text under 3, with comparitavely little under 3.1, 3.2 and 3.3).

- Figure 3: subplot labelling needs a bit more care

- 271: a single rooftop? The associated RF change in fig. 3b suggests this is a very large perturbation. On second reading I understand this is a localised and not global mean RF, it would be good to just confirm this in the text somewhere.

- 287: again can now update to 2020

- 296: 100 years (add unit)

- 333-336: I think this example nicely highlights the inherent danger of using metrics for policymaking without due understanding of the methods and assocaited uncertainties behind boiling everything down to one CO2-eq number.

- 452: aff/def: spell out

- 456: Smith et al 2020 published version now has 17 models

- Section 7: as section 3 (top heavy)

**References in this response not in manuscript already**

- Allen et al. 2018 https://www.nature.com/articles/s41612-018-0026-8

- Collins et al. 2020 https://iopscience.iop.org/article/10.1088/1748-9326/ab6039

- Jones et al. 2015 https://link.springer.com/article/10.1007/s10584-015-1411-5

- Nicholls et al. 2020a https://gmd.copernicus.org/articles/13/5175/2020/gmd-13-5175-2020.htm

- Nicholls et al. 2020b https://www.essoar.org/doi/10.1002/essoar.10504793.1

- Denison et al. 2019 https://iopscience.iop.org/article/10.1088/1748-9326/ab4df4

- Meinshausen et al. 2020 https://gmd.copernicus.org/articles/13/3571/2020/

- Jones et al. 2013 https://journals.ametsoc.org/view/journals/clim/26/13/jcli-d-12-00554.1.xml

---

## Referee Comment (RC2) · Anonymous Referee #2 · 26 Feb 2021

I found this review informative, and is a useful evaluation of options for CO2-equivalence methods for albedo change. Overall, as a review paper it naturally contains a fair amount of information. On the whole, it presents the information clearly, with a good structure and it is therefore straightforward to follow. I think it will be a useful paper to inform any potential future developments in how albedo change is evaluated in comparison to other changes to radiative forcing.

The paper highlights that there are difficulties with producing an equivalence between two fundamentally different forcings (a CO2 emission and a change to albedo), but that methods do exist, albeit with various drawbacks. It appears to me that scientific

accuracy is sacrificed for simplicity in the use of scalar metrics. The question for me is under what circumstances that renders the scalar metric useless. An example is shown which does this (e.g. around line 330, the choice of AF can lead you to different conclusions. Then the example in fig 6 also shows EESF is inappropriate.). Does this mean that EESF is inappropriate to use more generally, as the scientific representation is generally inaccurate? It seems to me that would be the case, and the only real benefit is because of simplicity/the status quo. I think a clear summary statement of whether you assess the scalar metrics to have any scientific benefit (when there are no technical hurdles to overcome, as there are for use by non-specialists). Or, the alternative is that the only benefit of a scalar metric is that it's easy to use for a non-specialist (which would imply to me that it's probably even easier to misuse. . .)

In addition to the methods you assess, I think GWP* would be worth assessing (or at least mentioning, if it's really not possible to assess). Lee et al (2021) evaluate the short-lived effects of aircraft, calculating CO2 equivalence to changes in RF using GWP*. The equation used in Lee et al is based on equations in Cain et al (2019) and Allen et al (2018). I think this would be highly relevant to your review paper, and I think it would be useful to evaluate where this method sits compared to the other methods, although I recognise it has only recently been published with respect to a radiative forcing. I think it ought to certainly be mentioned, and ideally discussed on its merits. Eg I think it should overcome the time dependency issues you have identified in other metrics.

CO2-forcing equivalent emissions (eg shown in Allen et al 2018, Jenkins et al 2018, Wigley 1998) are also another way to compare different RFs with one another, which may be worth considering/mentioning (although as this 'metric' uses a model, it's perhaps not strictly a metric).

I think the finding that the equivalence calculated using EESF is highly sensitive to AF is important, eg around line 330 you make this point with regard to policy/decision making, and your comments around line 500. That could be brought out in the abstract

as it highlights a key challenge for using this metric for albedo change as standard practice. This suggests to me that the other metrics are theoretically better suited for more general use (if not practically).

Specific comments

Line 50: worth pointing out that CO2 persists in the climate system for 100s to 1000s of years. If it was short-lived then the relationship would be different so I think it's important to mention this here, as it's why the CO2 emission is not reversible.

Line 78: 'The climate may respond differently to different perturbation types despite similar RF magnitudes, as feedbacks are not independent of the perturbation type' – do the two clauses in this sentence mean the same thing? If there is, remove the repetition. If there is a subtle difference, please make it explicit.

Fig 3a: alpha old line is too hard to see. The B) doesn't appear on panel B and D) is misplaced too.

Line 294: Can you explicitly explain why?

Fig 6 is very hard to follow. Can you make the explanation clearer and legend clearer? I am not sure if you discuss the red lines in fig 6a? Are the red lines simply the cumulative of the blue lines? If so, why is the red dashed line always +ve when the blue dashed line starts of -ve?

The example starting in line 400: you say that GWP is most appropriate here. However, if GWP works very well for GHGs like nitrous oxide, but not very well for albedo change and is subject to discrepancies that vary over time for albedo, then is it really appropriate? GWP works well for long lives gases as an equivalence metric for CO2. However where the impact varies over a shorter time period than CO2 (eg an albedo change scenario) then although the use of GWP could in some ways be seen as consistent, it is in some ways simply applying a metric that works for long lived gases to other forcings which are poorly suited. I can see GWP is useful because people already

use it. It doesn't really mean it's scientifically suitable. In summary, I'd suggest that for ease of use, GWP might be suitable here, but I believe that scientifically it will still be less suitable than TDEE. If that is correct, then I think it would be a useful distinction to make here.

Line 410-12: I think that the GWP* approach in Lee et al mentioned above could be mentioned here as well as the discussion, if you are unable to bring it in to the metrics analysed in the main part of the paper.

Line 503: Does the requirement of the use of a scalar metric defeat the purpose of using a metric for comparison for policy making /decision making? If you would make a different decision using a scalar and a vector metric, why even use the scalar metric at all, when the scalar metric pushes you into a different decision? (I am not sure how often the scalar metric would push you into a different decision – perhaps something for future work)

Line 549: This implies that using a model is more uncertain than using a metric. As the metrics are based on models, I do not see how this can be the case. Suggest wording this more carefully so as not to imply metrics are free of the model uncertainty, whereas they are based on those same models with their inherent uncertainty.

References

Allen, M. R., Shine, K. P., Fuglestvedt, J. S., Millar, R. J., Cain, M., Frame, D. J., & Macey, A. H. (2018). A solution to the misrepresentations of CO2-equivalent emissions of short-lived climate pollutants under ambitious mitigation. npj Climate and Atmospheric Science, 1(1), 16. https://doi.org/10.1038/s41612-018-0026-8

Cain, M., Lynch, J., Allen, M. R., Fuglestvedt, J. S., Frame, D. J., & Macey, A. H. (2019). Improved calculation of warming-equivalent emissions for short-lived climate pollutants. npj Climate and Atmospheric Science, 2(1), 29. https://doi.org/10.1038/s41612-019-0086-4

Jenkins, S., Millar, R. J., Leach, N., & Allen, M. R. (2018). Framing Climate Goals in Terms of Cumulative CO2-Forcing-Equivalent Emissions. Geophysical Research Letters, 45(6), 2795–2804. https://doi.org/10.1002/2017GL076173

Lee, D. S., Fahey, D. W., Skowron, A., Allen, M. R., Burkhardt, U., Chen, Q., . . . Wilcox, L. J. (2021). The contribution of global aviation to anthropogenic climate forcing for 2000 to 2018. Atmospheric Environment, 244(September 2020), 117834.

Wigley, T. M. L. (1998). The Kyoto Protocol: CO2, CH4 and climate implications. Geophysical Research Letters, 25(13), 2285–2288. https://doi.org/10.1029/98GL01855
* * *

---

## Author Comment (AC1) · 30 Mar 2021

**Author comments, ACP-2020-1109**

We thank both reviewers for their thoughtful and constructive commentary. Our responses to their comments including descriptions of actions taken to address them are provided below. Among the more notable revisions we implemented was the inclusion of a new section devoted entirely to the GWP* metric, including a quantitative assessment of its performance when applied to surface albedo change radiative forcings.

**Anonymous Referee #1**

This article provides a review of the literature on climate metrics to evaluate land use change as CO2-equivalent emissions. There are 6 main methods developed over the last 20 years to evaluate the CO2-equivalence of land use changes, of which GWP and sigma-TDEE are the two most recommended by the authors (though both have their own drawbacks). The majority if not all national emissions inventories under the Paris Agreement only take into account CO2-equivalent emissions for a basket of WMGHGs, and clearly there is an omission in the non-GHG forcing. AR5 reviewed metrics for SLCFs in section 8.7 (in any case they are not widely used) and this article highlights the lack of consistency for metrics for surface albedo changes (which are also seldom used in pracitce). In any case, metrics can not yet describe with acceptable accuracy the resulting climate impact of a particular future socioeconomic scenario (i.e. from an integrated assessment model; see Dension et al. 2019).

Thank you for the comment and reference. We share the opinion that additional metric research is needed, a conclusion that emerges in our review.

Main comment and recommendation

I am personally sceptical of the utility of metrics beyond well-mixed GHGs (and even then, with many problems) and I am not convinced that applying CO2-equivalance to land use changes is useful and robust. However, the authors are not necessarily claiming that it is, so the stated purpose of this paper - a review article on land-use metrics - is achieved. Additionally, practitioners in policymaking and life-cycle assessment applications, prone to using metrics without a full appreciation of where the numbers come from, are likely to continue to do so for the foreseeable future, so it is important that methods are critically reviewed and limitations communicated. I would recommend more discussion as to the limitations of metrics generally and landuse change specifically. Conceptually, the problem for land use is similar to assigning metrics to SLCFs; like land use change, radiative forcing from SLCFs is spatially heterogenous and theoretically (if not practically) reversible over short time scales, and like land use change many SLCFs co-vary with CO2 emissions. Several groups are actively trying to improve metrics, e.g. GWP* (Allen et al 2018) and CGWP and CGTP (Collins et al. 2020). These adjusted metrics have a better link to final climate impacts (in terms of global mean surface temperature changes) and could include the effects of non-GHG forcers (through the CO2-forcing equivlant measure in Allen et al. 2018), though at the risk and expense of introducing additional complexity. Another way in which land use metrics are difficult to apply is that RF from land use change is small and uncertain (the authors do discuss this). Metrics for GHGs work well enough because the emissions to concentrations to radiative forcing (to temperature, in GTP-style metrics) relationships are well-defined, invertible (at least up until Etminan) and known within acceptable uncertainty. The alternative to metrics is to use a simple climate model to determine impacts. The two most recommended methods in this article, sigma-TDEE and GWP, need estimates of the radiative forcing of land use change. Therefore, why not cut out the middle man: calculate RF from land use change, calculate or estimate the change in land-use related CO2 sources or sinks for

afforestation/deforestation (if that is the intervention of interest, for a geoengineering-style experiment, the land-use CO2 does not need to be perturbed), and run these scenarios in a simple climate model that evaluates the emissions - concentrations - radiative forcing - temperature process, and determine the temperature difference between this and a reference scenario. Several candidate simple climate models are discussed in Nicholls et al. (2020a, 2020b). The required RF estimate can be obtained from the kernel method that the authors describe, or from something like Jones et al. (2015).

We appreciate that the reviewer sees some value in our work. As imperfect as they are, metrics will in all likelihood continue to be applied (as stated by the reviewer), and hence our job as researchers ought to be to point out the conditions or contexts in which they have most merit in addition to those in which they have little to no merit. As acknowledged by the reviewer – this was achieved in our review. We acknowledge, however, that their limitations were perhaps under communicated, thus we elected to expand the Discussion (new Section 8.1 and revised Section 8.3) to include more text surrounding the limitations or caveats of metrics in general – and specifically those applied in land use/land cover change contexts.

We also acknowledge that the GWP* metric was absent and deserved attention in our review – particularly in the context of surface albedo change – thus we elected to add a new section devoted entirely to it (new Section 6). To our knowledge this is the first effort to review this metric in the context of surface albedo change – a contribution we feel adds novelty and enhances overall value.

As to the comment surrounding the merits of simple-/reduced complexity climate models over the reviewed metrics, we have difficulties seeing the relevancy of introducing such a discussion here as the scope of our review is limited to RF-based measures, which simplified models do not do (even though the temperature response is arguably a more useful indicator of the climate impact).

Line-by-line comments

• 13: equivalent Corrected.

• 24: Earth (capital E) There are two occurrences of the word in this sentence. The "E" of the first is capitalized because the word is used as a proper noun (i.e., as a specific place); the "e" of the second remains lowercase because the word is used as a common noun ("the earth").

• 26: the 13 We are unsure about what needs correction here…

• 30: "large scale carbon dioxide removal" - mention afforestation specifically here, and why A good suggestion which we have executed, thanks.

• 38: "backward looking measure" - explain what is meant here. RF is not necessarily backward looking, and several future projections exist (including for radiative forcing from land use changes) We acknowledge that this sentence was poorly formulated. We meant that one can derive projections of RF, but the RF would still be relative to some past time period (i.e., pre-industrial, present day, etc.). We have revised and clarified.

• 45: UNFCCC We are unsure about what needs correction here…

• 47: "debate about GWP as the metric of choice" - perhaps add a citation here. Almost anything from Myles Allen's group 2018-2020 would fit the bill, or Denison et al 2019 We added a citation to the Denison et al. 2019 paper – thanks.

• 58: AR4 Corrected.

• 69-70: True up until AR5, where metrics were only done in rigorousness for WMGHGs and there was no evidence to suggest that ERF differed from RF. I would think that ERF would be preferred for metric calculations now, if it differed from RF. Of course we wait for AR6 for this to be defined, though might be worth a mention to future-proof this paper a little. Good idea about "future-proofing", which we have now done along the lines suggested – thanks.

• 77-85: This paragraph implies why ERF is a better idea than RF; by including tropospheric and land surface adjustments, temperature responses track ERF more closely than RF, and reduces (perhaps elimiates) the need for efficacy scalings.

• 88: is it definitely regression? Replaced "regression fits" with "curve fits".

• 90: this is outdated: use Etminan et al. 2016 or the re-fitted Etminan relationship in Meinshausen et al. 2020. We are aware of this but chose to document the reference gas state-of-the-art at the time of AR5 (i.e., Myhre et al. 2013) as it will remain the prevailing convention for metric calculations until AR6. With this in mind we felt it made better pedagogic sense to introduce Etminan et al. 2016 at line 99. As the reviewer acknowledges two comments later, the effect of the Etminan update in terms of a $CO_2$ doubling is small (ca. 2%), and a 1 ppm change around the present day does not affect CO2's radiative efficiency much.

• 94-96: the dependency on background state of the reference gas highlights a weakness of metrics generally; more should be made of this. We agree that is a weakness of metrics in general, but find it out of scope to go more in-depth here, as there are numerous papers dedicated to this topic.

• 99-101: the Etminan relationship has even modified the RF from 2xCO2 (3.80 W/m2 versus 3.71 in Myhre et al. 1998) but the point here is that a 1ppm change around present day doesn't affect the radiative efficiency of CO2 much, I guess.

• 116: y(t) could be generalised and could represent individual carbon cycle models, as well as the multi-model mean as was used in Myhre et al. 2013 Fair enough, we have revised and generalized.

• 121-123: you could explain why this is important for metrics and why metric calculations based on AR5 values might be incorrect under different future assumptions (including for 1.5C targets, where the impulse response function between pre-industrial and present-day states are substantially different as shown by Millar et al. 2017) We believe we have drawn sufficient attention to the issue given the core focus of the paper on metrics for the non-reference gas. Interested readers can visit the Millar et al. 2017 paper which we have referenced here.

• 139: Block and Mauritsen reference should be 2013, not 2015 Thanks for spotting this. We had referenced the source of the kernels, not the original article for which they were derived (the final version of which was published in 2014). The same was done for the Smith and Pendergrass kernels – we have updated these references as well, pointing to the original papers and not the datasets.

• 153-154: To use eq. 5 don't you need climate model output? I suppose you could do this based on observations but it won't be globally complete and will still need a fair amount of data crunching. Maybe I've misunderstood something. Eq. (5) needs downward solar radiation incident at surface and TOA, the former of which can either be measured in-situ (with upward-looking radiometers) or based on optical satellite remote sensing (e.g., CERES EBAF or NASA SRB).

• table 3: Sciusco et al: should it be "on" rather than "no"? Corrected.

• 177-179: here is another demonstrable weakness of attempting to use a $CO_2$-eq metric for land use change. While AF can be assumed to be = 0.47 in the recent past, there's no guarantee this would hold in the future (Millar et al. (2017), and Jones et al (2013) on AF in RCPs). We agree and have pointed this weakness/uncertainty out on several occasions throughout the manuscript.

• Figure 1: could now be updated to 2019 using the latest GCP update We feel there is no appreciable benefit of updating this figure with one additional data point (which was not available at the time of submission).

• 191-192: "approximate time frame..." this half of the sentence doesn't explain what's going on here - does it mean that 47 It appears that a part of this comment is missing…

• 238: is the minus sign meant to be there after TH? We're unsure what the reviewer is referring to here…

Section 3 subsectioning: it's rather top-heavy (most of the text under 3, with comparitavely little under 3.1, 3.2 and 3.3). We feel this section is logically structured as it is. Does the reviewer have a suggestion for how Section 3 might be sub-sectioned in a more logical or clear way? We note that Anonymous Reviewer #2 takes no issue with the structure or clarity of our presentation.

• Figure 3: subplot labelling needs a bit more care Added the missing panel B)'s label – thanks.

• 271: a single rooftop? The associated RF change in fig. 3b suggests this is a very large perturbation. On second reading I understand this is a localised and not global mean RF, it would be good to just confirm this in the text somewhere. We agree that the additional clarity was needed to better contextualize results presented in this figure. We have therefore made it explicit in the preceding text and in the figure's caption that results correspond to 1 m2 of perturbed rooftop and associated *local* RF.

• 287: again can now update to 2020 Again we feel this effort is not justified given the intent of this section (and paper more generally) of simply demonstrating the sensitivity of metric values to the choice of AF.

• 296: 100 years (add unit) Unit added.

• 333-336: I think this example nicely highlights the inherent danger of using metrics for policymaking without due understanding of the methods and assocaited uncertainties behind boiling everything down to one $CO_2$-eq number. Thanks. This is the type of knowledge we hope our work can contribute to increasing among policy makers.

• 452: aff/def: spell out Corrected.

• 456: Smith et al 2020 published version now has 17 models Corrected.

• Section 7: as section 3 (top heavy) We have revised this section (now Section 8) and added a new subsection heading – the result of which now provides greater balance.

References in this response not in manuscript already

• Allen et al. 2018 https://www.nature.com/articles/s41612-018-0026-8

• Collins et al. 2020 https://iopscience.iop.org/article/10.1088/1748-9326/ab6039

• Jones et al. 2015 https://link.springer.com/article/10.1007/s10584-015-1411-5

• Nicholls et al. 2020a https://gmd.copernicus.org/articles/13/5175/2020/gmd-13- 5175-2020.htm

• Nicholls et al. 2020b https://www.essoar.org/doi/10.1002/essoar.10504793.1

• Denison et al. 2019 https://iopscience.iop.org/article/10.1088/1748-9326/ab4df4

• Meinshausen et al. 2020 https://gmd.copernicus.org/articles/13/3571/2020/

• Jones et al. 2013 https://journals.ametsoc.org/view/journals/clim/26/13/jcli-d-12- 00554.1.xml

**Anonymous Referee #2**

I found this review informative, and is a useful evaluation of options for CO2- equivalence methods for albedo change. Overall, as a review paper it naturally contains a fair amount of information. On the whole, it presents the information clearly, with a good structure and it is therefore straightforward to follow. I think it will be a useful paper to inform any potential future developments in how albedo change is evaluated in comparison to other changes to radiative forcing. The paper highlights that there are difficulties with producing an equivalence between two fundamentally different forcings (a CO2 emission and a change to albedo), but that methods do exist, albeit with various drawbacks. We appreciate that the reviewer finds value in our review – both in terms of its content and presentation.

It appears to me that scientific accuracy is sacrificed for simplicity in the use of scalar metrics. The question for me is under what circumstances that renders the scalar metric useless. An example is shown which does this (e.g. around line 330, the choice of AF can lead you to different conclusions. Then the example in fig 6 also shows EESF is inappropriate.). Does this mean that EESF is inappropriate to use more generally, as the scientific representation is generally inaccurate? It seems to me that would be the case, and the only real benefit is because of simplicity/the status quo. I think a clear summary statement of whether you assess the scalar metrics to have any scientific benefit (when there are no technical hurdles to overcome, as there are for use by non-specialists). Given our decision to incorporate GWP* into the review (see next reply), we no longer see any context in which the EESF metric has merit. We state this explicitly in our revised Discussion (new section 8.1). Or, the alternative is that the only benefit of a scalar metric is that it's easy to use for a nonspecialist (which would imply to me that it's probably even easier to misuse. . .) In addition to the methods you assess, I think GWP* would be worth assessing (or at least mentioning, if it's really not possible to assess). Reviewer #1 shares this assessment and we agree. We have therefore devoted an entirely new section to the GWP* metric (new Section 6) – with due citations to Lee et al. (2021) and other important works. Lee et al (2021) evaluate the short-lived effects of aircraft, calculating CO2 equivalence to changes in RF using GWP*. The equation used in Lee et al is based on equations in Cain et al (2019) and Allen et al (2018). I think this would be highly relevant to your review paper, and I think it would be useful to evaluate where this method sits compared to the other methods, although I recognise it has only recently been published with respect to a radiative forcing. We agree and add a quantitative evaluation of its merits as a metric for surface albedo change, comparing it to the TDEE metric (new Section 6). We also qualitatively discuss its merits relative to the EESF measure, both in (new) Section 6 and (new) Section 8.1. I think it ought to certainly be mentioned, and ideally discussed on its merits. See above. Eg I think it should overcome the time dependency issues you have identified in other metrics. Indeed it does, which we now show quantitatively in (new) Figure 7B, state quantitatively in the surrounding text, and re-iterate in the Discussion. CO2-forcing equivalent emissions (eg shown in Allen et al 2018, Jenkins et al 2018, Wigley 1998) are also another way to compare different RFs with one another, which may be worth considering/mentioning (although as this 'metric' uses a model, it's perhaps not strictly a metric). We appreciate the reviewer

drawing our attention to the works of Wigley (1998) and the "CO2-forcing-equivalence" approach of Jenkins et al. (2018), the latter being conceptually similar to the TDEE approach. We now mention this in Section 3 and provide citations to these works and others. I think the finding that the equivalence calculated using EESF is highly sensitive to AF is important, eg around line 330 you make this point with regard to policy/decision making, and your comments around line 500. That could be brought out in the abstract as it highlights a key challenge for using this metric for albedo change as standard practice. This suggests to me that the other metrics are theoretically better suited for more general use (if not practically). We believe the level of detail of the current abstract is well balanced and do not share the opinion that the EESF metric should be singled out and made visible here. An interested reader will arrive at this conclusion after reading the revised Discussion.

Specific comments

Line 50: worth pointing out that CO2 persists in the climate system for 100s to 1000s of years. If it was short-lived then the relationship would be different so I think it's important to mention this here, as it's why the CO2 emission is not reversible. Revised – we now explicitly state the time scale involved ("millennia").

Line 78: 'The climate may respond differently to different perturbation types despite similar RF magnitudes, as feedbacks are not independent of the perturbation type' – do the two clauses in this sentence mean the same thing? If there is, remove the repetition. If there is a subtle difference, please make it explicit. Yes, they do say the same thing. We have re-phrased this sentence to make this clearer.

Fig 3a: alpha old line is too hard to see. The B) doesn't appear on panel B and D) is misplaced too. Corrected.

 Line 294: Can you explicitly explain why? Fair enough – we provide explanations following the statements made both in this sentence and the succeeding.

Fig 6 is very hard to follow. Can you make the explanation clearer and legend clearer? I am not sure if you discuss the red lines in fig 6a? Are the red lines simply the cumulative of the blue lines? If so, why is the red dashed line always +ve when the blue dashed line starts of -ve? We have provided additional description in the figure's caption. Descriptions of the red lines are provided in the surrounding text, including enhanced legend descriptions. The dashed red line in 6a is CO2's AGWP and is unrelated to any of the blue curves.

The example starting in line 400: you say that GWP is most appropriate here. However, if GWP works very well for GHGs like nitrous oxide, but not very well for albedo change and is subject to discrepancies that vary over time for albedo, then is it really appropriate? GWP works well for long lives gases as an equivalence metric for CO2. However where the impact varies over a shorter time period than CO2 (eg an albedo change scenario) then although the use of GWP could in some ways be seen as consistent, it is in some ways simply applying a metric that works for long lived gases to other forcings which are poorly suited. I can see GWP is useful because people already use it. It doesn't really mean it's scientifically suitable. In summary, I'd suggest that for ease of use, GWP might be suitable here, but I believe that scientifically it will still be less suitable than TDEE. If that is correct, then I think it would be a useful distinction to make here. We appreciate this comment and have revised this section along the lines suggested, stating explicitly that GWP has little scientific merit when applied to assess short-lived forcings. This is supported by a a new quantitative comparison of GWP and ΣTDEE in (new) Section 8.1 and (new) Table 6.

Line 410-12: I think that the GWP* approach in Lee et al mentioned above could be mentioned here as well as the discussion, if you are unable to bring it in to the metrics analysed in the main part of the paper.  We agree that this is the most natural place in the manuscript to introduce the GWP* metric and have thus chosen to insert our new GWP* section (new Section 6) here.

Line 503: Does the requirement of the use of a scalar metric defeat the purpose of using a metric for comparison for policy making /decision making? If you would make a different decision using a scalar and a vector metric, why even use the scalar metric at all, when the scalar metric pushes you into a different decision? (I am not sure how often the scalar metric would push you into a different decision – perhaps something for future work) We appreciate the logic of this comment – that there is often a trade-off between simplicity (or applicability) and scientific integrity when it comes to scalar metrics.  We agree that the desirability of scalar metrics in decision-making can be self-defeating, which is why we had stated (in the same sentence) that vector or "time series-based" metrics have greatest merit.  Because scalar metrics are omnipresent and unlikely to disappear in the near-term due to scientific concerns about their integrity, we chose to frame their merits in the decision-support context rather than dismiss or criticize them outright (which we felt was reaching a bit outside the manuscript scope).

Line 549: This implies that using a model is more uncertain than using a metric. As the metrics are based on models, I do not see how this can be the case. Suggest wording this more carefully so as not to imply metrics are free of the model uncertainty, whereas they are based on those same models with their inherent uncertainty.  Fair point!  This sentence was indeed poorly formulated and has been completely overhauled.

**References**

Allen, M. R., Shine, K. P., Fuglestvedt, J. S., Millar, R. J., Cain, M., Frame, D. J., & Macey, A. H. (2018). A solution to the misrepresentations of CO2-equivalent emissions of short-lived climate pollutants under ambitious mitigation. npj Climate and Atmospheric Science, 1(1), 16. https://doi.org/10.1038/s41612-018-0026-8

Cain, M., Lynch, J., Allen, M. R., Fuglestvedt, J. S., Frame, D. J., & Macey, A. H. (2019). Improved calculation of warming-equivalent emissions for short-lived climate pollutants. npj Climate and Atmospheric Science, 2(1), 29. https://doi.org/10.1038/s41612- 019-0086-4

Jenkins, S., Millar, R. J., Leach, N., & Allen, M. R. (2018). Framing Climate Goals in Terms of Cumulative CO2-Forcing-Equivalent Emissions. Geophysical Research Letters, 45(6), 2795–2804. https://doi.org/10.1002/2017GL076173

Lee, D. S., Fahey, D. W., Skowron, A., Allen, M. R., Burkhardt, U., Chen, Q., . . . Wilcox, L. J. (2021). The contribution of global aviation to anthropogenic climate forcing for 2000 to 2018. Atmospheric Environment, 244(September 2020), 117834.

Wigley, T. M. L. (1998). The Kyoto Protocol: CO2, CH4 and climate implications. Geophysical Research Letters, 25(13), 2285–2288. https://doi.org/10.1029/98GL01855

---

## Referee Report (RR1)

Review Bright and Lund

Thanks to the authors for addressing all the reviewer comments. I have a few additional comments, which relate to new material which has been introduced at the suggestion of the reviewers. Subject to these new comments being addressed, I am happy for the manuscript to be accepted.

Figure 6:
I still think this figure is a bit confusing. The main text around line 353 says we have a new example where we harvest a broadleaf forest and plant an evergreen forest. This gives you the profile of RF in the solid blue line in fig 6. That is clear. However, the introduction of the red curves in the text is less clear. The dashed red is the sum of the RF over time after a 1kg pulse emissions of CO2. What exactly is the solid red line, i.e. what change in albedo is causing it? Can you explain how the two red curves relate to each other / why you are including them?

Section 6:
The equation for GWP* in Lee et al is the version from Cain et al (2019), but Lee et al say that (what they call) alpha is assumed to be zero for their case. Where you apply GWP* in fig 7, as you have a full time series, I think you can assume that alpha is not zero. You could then use the full equation which accounts for the average RF over the period Delta-t. As this accounts for the slower climate response to past changes to RF, perhaps GWP* will have better agreement to Delta T in fig 7b. The equation in Cain et al tried to improve on the Allen et al 2018 equation to have a better agreement with temperature, so it may do so in your example and I think it's worth testing. If that isn't possible, then I think you need to say that you haven't used the extra term in Cain et al (and why) and discuss whether you think it would improve the agreement with temperature (or not). You may also want to then amend your discussion around line 581 related to GWP*.

Regarding the choice of time horizon in GWP* - the authors of GWP* use H=100 years and say that:

'In defining CO2-e and CO2-e* emissions, we use $H$=100 years following established practice. Results under GWP* are insensitive to this provided $H$ is much greater than the lifetime of the SLCP because the absolute GWP of an SLCP becomes a constant at these timescales, while the AGWP$H$ of the reference gas, CO2, increases linearly with $H$—see ref. 3 and Fig. 8.29 of ref. 14 Hence the $H$-dependence cancels out in the calculation of CO2-e* for both SCLP emissions and radiative forcing. In contrast, GWP-based CO2-e values for SLCPs scale approximately with $1/H$, making the nominal relative importance of SLCPs and cumulative pollutants acutely sensitive to this choice of time-horizon.' (Allen et al 2018)

So I don't think that GWP* uses a subjective choice of time horizon like standard GWP100 does, as you have said around line 447, and suggest that you discuss what is in the paragraph I have quoted instead.

---

## Author Response (AR2)

Thanks to the authors for addressing all the reviewer comments. I have a few additional comments, which relate to new material which has been introduced at the suggestion of the reviewers. Subject to these new comments being addressed, I am happy for the manuscript to be accepted.

Figure 6:
I still think this figure is a bit confusing. The main text around line 353 says we have a new example where we harvest a broadleaf forest and plant an evergreen forest. This gives you the profile of RF in the solid blue line in fig 6. That is clear. However, the introduction of the red curves in the text is less clear. The dashed red is the sum of the RF over time after a 1kg pulse emissions of CO2. What exactly is the solid red line, i.e. what change in albedo is causing it? Can you explain how the two red curves relate to each other / why you are including them?

OK, we have provided additional explanation in Figure 6's caption about the two red curves plotted in panel A and why they are shown.

Section 6:
The equation for GWP* in Lee et al is the version from Cain et al (2019), but Lee et al say that (what they call) alpha is assumed to be zero for their case. Where you apply GWP* in fig 7, as you have a full time series, I think you can assume that alpha is not zero. You could then use the full equation which accounts for the average RF over the period Delta-t. As this accounts for the slower climate response to past changes to RF, perhaps GWP* will have better agreement to Delta T in fig 7b. The equation in Cain et al tried to improve on the Allen et al 2018 equation to have a better agreement with temperature, so it may do so in your example and I think it's worth testing. If that isn't possible, then I think you need to say that you haven't used the extra term in Cain et al (and why) and discuss whether you think it would improve the agreement with temperature (or not). You may also want to then amend your discussion around line 581 related to GWP*.

OK, we have invested notable effort here to demonstrating the faithfulness by which the GWP* approach reproduces the temperature response (revised Figure 7) for a range of time step sizes ("Delta-t") and "alpha" factors applied to the same widely divergent RF scenarios as used in the previous version of Figure 7 A.  Although not easy, we believe we have been able to strike a good balance between adding new content which serves to further elevate the manuscript's scientific value while maintaining an orderly and logical flow.  We feel that any additional elaboration on the GWP* measure at this point would begin to extend well beyond the current manuscript scoping.

Regarding the choice of time horizon in GWP* - the authors of GWP* use H=100 years and say that:

'In defining CO2-e and CO2-e* emissions, we use $H$=100 years following established practice. Results under GWP* are insensitive to this provided $H$ is much greater than the lifetime of the SLCP because the absolute GWP of an SLCP becomes a constant at these timescales, while the AGWP$H$ of the reference gas, CO2, increases linearly with $H$—see ref. 3 and Fig. 8.29 of ref. 14 Hence the $H$-dependence cancels out in the calculation of CO2-e* for both SCLP emissions and radiative forcing. In contrast, GWP-based CO2-e values for SLCPs scale approximately with $1/H$, making the nominal relative importance of SLCPs and cumulative pollutants acutely sensitive to this choice of time-horizon.' (Allen et al 2018)

So I don't think that GWP* uses a subjective choice of time horizon like standard GWP100 does, as you have said around line 447, and suggest that you discuss what is in the paragraph I have quoted instead.

This is a fair comment and we have now revised the content as suggested and expanded Section 6 to provide a more up-to-date and balanced review of the state of the GWP* measure.

---

## Author Response (AR3)

**Editor comment**: Thank you very much for addressing the points raised in the previous review. I also accept that you do not want to go into much more detail regarding GWP*. However, could you please clarify the question about the usage of alpha (from Cain/Lee) to ensure full reproducibility and traceability to the references?

**Author reply to Editor comment:**

We appreciate the comment about the need for improved clarity surrounding our usage of the weighting factor "alpha" in our last revision where we had elected to accommodate the reviewer's suggestion (pasted below for convenience) to assess whether better agreement with $\Delta T$ ("Delta T") could be achieved when applying the modified GWP* expression presented as Eq. 1 in Lee et al., 2020.  Because we had used the notation "alpha" extensively throughout our manuscript to denote the surface albedo, we elected instead to use the original "s" notation of Cain et al., 2019 (the original presentation of the modified GWP* expression) to avoid confusion.

We decided to reproduce Eq. 1 of Lee et al. 2020 for the reader's convenience, now appearing as new equation "10" in our newly revised manuscript.  We now state explicitly in the main text and in Fig. 7's caption that Figures 7 B-D are based on this equation, and that the "alpha" and "s" terms used in Lee et al. 2019 and Cain et al. 2020 have identical interpretations.

**Reviewer comment, previous review:**

Section 6:
The equation for GWP* in Lee et al is the version from Cain et al (2019), but Lee et al say that (what they call) alpha is assumed to be zero for their case. Where you apply GWP* in fig 7, as you have a full time series, I think you can assume that alpha is not zero. You could then use the full equation which accounts for the average RF over the period Delta-t. As this accounts for the slower climate response to past changes to RF, perhaps GWP* will have better agreement to Delta T in fig 7b. The equation in Cain et al tried to improve on the Allen et al 2018 equation to have a better agreement with temperature, so it may do so in your example and I think it's worth testing. If that isn't possible, then I think you need to say that you haven't used the extra term in Cain et al (and why) and discuss whether you think it would improve the agreement with temperature (or not). You may also want to then amend your discussion around line 581 related to GWP*.

**Author reply to Reviewer comment, previous review:**

OK, we have invested notable effort here to demonstrating the faithfulness by which the GWP* approach reproduces the temperature response (revised Figure 7) for a range of time step sizes ("Delta-t") and "alpha" factors applied to the same widely divergent RF scenarios as used in the previous version of Figure 7 A.  Although not easy, we believe we have been able to strike a good balance between adding new content which serves to further elevate the manuscript's scientific value while maintaining an orderly and logical flow.  We feel that any additional elaboration on the GWP* measure at this point would begin to extend well beyond the current manuscript scoping.